# Controlled Sparsity via Constrained Optimization or:
## How I Learned to Stop Tuning Penalties and Love Constraints

Jose Gallego-Posada[*]    Juan Ramirez    Akram Erraqabi

Yoshua Bengio[‡]    Simon Lacoste-Julien[‡]

Mila and DIRO, Université de Montréal, Canada

[‡] Canada CIFAR AI Chair

## Abstract

The performance of trained neural networks is robust to harsh levels of pruning. Coupled with the ever-growing size of deep learning models, this observation has motivated extensive research on learning sparse models. In this work, we focus on the task of controlling the level of sparsity when performing sparse learning. Existing methods based on sparsity-inducing penalties involve expensive trial-and-error tuning of the penalty factor, thus lacking direct control of the resulting model sparsity. In response, we adopt a *constrained* formulation: using the gate mechanism proposed by Louizos et al. [31], we formulate a constrained optimization problem where sparsification is guided by the training objective and the desired sparsity target in an end-to-end fashion. Experiments on CIFAR-{10, 100}, Tiny-ImageNet, and ImageNet using WideResNet and ResNet{18, 50} models validate the effectiveness of our proposal and demonstrate that we can reliably achieve pre-determined sparsity targets without compromising on predictive performance.

## 1 Introduction

Commonly used neural networks result in *overparametrized* models, whose performance is robust to harsh levels of parameter pruning [18, 42, 12, 14]. Thus, regularization techniques aimed at learning sparse models can drastically reduce the computational cost associated with the learnt model by removing unnecessary parameters, and retain good performance in the learning task. Given the recent research trends which explore the capabilities of ever more ambitious large-scale models [2], developing techniques which provide reliable training of *sparsified* models becomes crucial for deploying them in massively-used systems, or on resource-constrained devices.

Pruning methods aim to reduce the storage and/or computational footprint of a model by discarding individual parameters [18, 34] or groups thereof [29, 27, 37], while inducing minimal distortion in the model's predictions. These methods can be further categorized based on whether the sparse model is obtained *while* or *after* training the model (also known as *in-training* and *post-training* sparsification).

Traditional post-training methods rely on heuristic rankings of the weights or filters to be pruned, often based on parameter magnitudes [26, 18]. Despite their simplicity, these methods usually require retraining the weights to maintain high accuracy after pruning, and thus incur in additional computational overhead. On the other hand, in-training methods which *learn* a good sparsity pattern by augmenting the training loss with sparsity-inducing penalties [31, 27] do not perform fine-tuning, but face challenges regarding the tuning and interpretability of the penalty hyperparameter.

In this work[†], we focus on the task of learning models with *controlled* levels of sparsity while performing in-training pruning. We tackle two central issues of the popular penalized method of Louizos et al. [31]: ① tuning the $L_0$-penalty coefficient to achieve a desired sparsity level is non trivial and can involve computationally wasteful trial-and-error attempts; ② in the worst case the penalized method can outright fail at producing any sparsity, as documented by Gale et al. [14].

---

[*]Correspondence to: {gallegoj, juan.ramirez, akram.erraqabi}@mila.quebec

[†]Our code is available at: https://github.com/gallego-posada/constrained_sparsity

36th Conference on Neural Information Processing Systems (NeurIPS 2022).

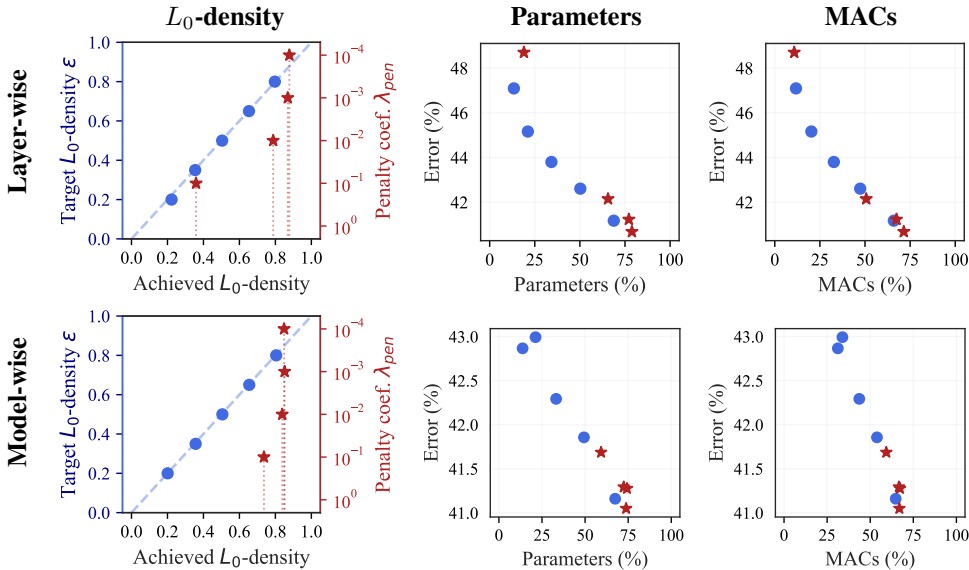

Figure 1: Training sparse ResNet18 models on TinyImageNet [28]. Density denotes the proportion of active gates in the model. The penalty-based method (red) shows a stagnating-then-overshooting behavior, making it difficult to tune. In contrast, our proposed constrained approach (blue) reliably achieves the desired target $L_0$-densities. The diagonal denotes the ideal case in which the achieved density exactly matches the target density used in the constrained setting. Parameters and MACs are computed for the corresponding test-time *purged* networks following the procedure described in Appendix D; the $L_0$-density (see Eq. (3)) is computed for the train-time model.

To address these limitations, we propose a constrained optimization approach in which ***arbitrary sparsity targets are expressed as constraints*** on the $L_0$-norm of the parameters. Formally, we consider constraints of the type $\|\boldsymbol{\theta}_g\|_0 \leq K$, where $\boldsymbol{\theta}_g$ represents a group $g$ of parameters of the network (e.g. individual layers, or the whole model), and resort to well established gradient-based methods for optimizing the Lagrangian associated with the constrained optimization problem.

Adopting this constrained formulation provides several advantages:

- Unlike the multiplicative factor $\lambda$ of a penalty term, the constraint level $\epsilon_g$ has straightforward and **interpretable semantics** associated with the *density* of a block of parameters $\boldsymbol{\theta}_g$, i.e. the percentage of active parameters.
- Requiring different density levels for different parameter groups (e.g. lower density for network modules with a larger computational or memory footprint), simply amounts to specifying several constraints with levels matching these desired densities, thus **avoiding the costly process of trial-and-error tuning**[‡] and re-balancing various penalty factors.
- Much like the penalized approach in which additional regularizers can be "stacked" as other additive terms in the objective, new desired properties can be expressed in the constrained formulation in a **modular and extensible** fashion as additional constraints.
- In non-convex problems, the constrained formulation **can be strictly more powerful** than the penalized approach: there may be constraint levels that *cannot be achieved by any value of the penalty coefficient* [1, §4.7.4].

The left column of Fig. 1 illustrates the interpretability and controllability advantages of the constrained approach when training a sparse ResNet18 model on TinyImageNet. We vary the constraint level (left axis) and the penalty coefficient (right axis) and compare the achieved parameter density at the end of training. Note how the penalized approach results in an very dense model ($> 80\%$) across several orders of magnitude of the penalty factor, and then suddenly drops to $< 40\%$ density. This behavior is in stark contrast with our proposed constrained approach, which *consistently achieves the desired target density*, across a wide range of values. See Section 5 for further discussion.

The purpose of our paper is to illustrate the feasibility and advantages of using constrained formulations in the study of sparse learning. We favor Lagrangian, gradient-based methods for tackling the constrained optimization problem due to their ease of use and scalability in the context of machine learning models. Exploring alternative constrained optimization techniques is an interesting direction for future studies, but lies beyond the scope of our work.

---

[‡]Appendix E shows that the tuning challenges of penalized methods exist even for simple MLP tasks.

The main contributions of this work are:

- Building on the work of Louizos et al. [31], we propose a constrained approach for learning models with controllable levels of sparsity (Section 3).
- We introduce a *dual restart* heuristic to avoid the excessive regularization caused by the accumulation of constraint violations in gradient-based Lagrangian optimization (Section 3).
- Previous studies [31, 14] have been unsuccessful at training sparse ResNets [45] based on $L_0$ regularization without significantly damaging performance. We propose two simple adjustments to the implementation of Louizos et al. [31], allowing us to overcome these challenges (Appendix H).
- We provide empirical evidence that we can reliably achieve controllable sparsity across many different architectures and datasets. Moreover, the controlability and interpretability benefits of the constrained approach do not come at the expense of achieving competitive predictive performance (Section 5).

## 2 Sparsity via $L_0$ Penalties

Louizos et al. [31] propose a framework for learning sparse models using the $L_0$-"norm" of the model parameters as an additive penalty to the usual training objective. The $L_0$-norm counts the *number* of non-zero entries in the parameter vector, and ignores the magnitude of said entries. Consider $h(x; \boldsymbol{\theta})$ be a predictor with parameters $\boldsymbol{\theta}$ and a supervised learning problem defined by a dataset of $N$ i.i.d. pairs $\mathcal{D} = \{(x_i, y_i)\}_{i=1}^N$, a loss function $\ell$ and a regularization coefficient $\lambda_{\text{pen}} \geq 0$. Louizos et al. [31] formulate the $L_0$-regularized empirical risk objective:

$$\mathcal{R}(\boldsymbol{\theta}, \lambda_{\text{pen}}) = \mathcal{L}_{\mathcal{D}}(\boldsymbol{\theta}) + \lambda_{\text{pen}}\|\boldsymbol{\theta}\|_0 = \frac{1}{N}\left(\sum_{i=1}^N \ell\left(h(x_i; \boldsymbol{\theta}), y_i\right)\right) + \lambda_{\text{pen}}\sum_{j=1}^{|\boldsymbol{\theta}|} \mathbb{1}\{\theta_j \neq 0\} \quad (1)$$

The non-differentiability of the $L_0$-norm makes it poorly suited for gradient-based optimization. The authors propose a reparametrization $\boldsymbol{\theta} = \tilde{\boldsymbol{\theta}} \odot \mathbf{z}$, where $\tilde{\boldsymbol{\theta}}$ are free (signed) parameter magnitudes, and $\mathbf{z}$ are independent stochastic *gates* indicating whether a parameter is active[§]. The authors model the gates using a modified version of the concrete distribution [32, 22], with parameters denoted by $\boldsymbol{\phi}$.

This reparametrization allows for gradient-based optimization procedures, while retaining the possibility of achieving *exact* zeros in the parameters values. We provide a brief overview of the properties of the concrete distribution in Appendix A.

Moreover, this stochastic reparametrization induces a distribution over the network parameters $\boldsymbol{\theta}$. In consequence, the authors propose to re-define the training objective as the expectation (under the distribution of the gates) of the $L_0$-regularized empirical risk in Eq. (1):

$$\mathcal{R}(\tilde{\boldsymbol{\theta}}, \boldsymbol{\phi}, \lambda_{\text{pen}}) \triangleq \mathbb{E}_{\mathbf{z} \mid \boldsymbol{\phi}}\left[\mathcal{R}(\tilde{\boldsymbol{\theta}} \odot \mathbf{z}, \lambda_{\text{pen}})\right] = \mathbb{E}_{\mathbf{z} \mid \boldsymbol{\phi}}\left[\mathcal{L}_{\mathcal{D}}(\tilde{\boldsymbol{\theta}} \odot \mathbf{z})\right] + \lambda_{\text{pen}}\mathbb{E}_{\mathbf{z} \mid \boldsymbol{\phi}}\left[\|\mathbf{z}\|_0\right] \quad (2)$$

**Test-time model.** Since the stochastic reparametrization induces a distribution over models, Louizos et al. [31] propose a protocol to choose a sparse network at test time. We employ a slightly modified version of their strategy, based on the *medians* of the gates (see Appendix A.1).

**Parameter grouping.** Rather than considering a gate for each individual parameter (which would double the number of trainable parameters), several parameters may be gathered under a shared gate. We match the setup of Louizos et al. [31] who focus on *neuron sparsity*: using ① one gate per *input neuron* for fully connected layers; and ② one gate per *output feature map* for convolutional layers. This use of *structured sparsity* results in practical storage and computation improvements since entire parameter groups (e.g. slice of convolution kernels/activation) can be discarded.

**Combining other norms.** Louizos et al. [31] show that their reparametrization can used in conjunction with other commonly used norms for regularization, such as the $L_2$-norm. One can express $\mathbb{E}_{\mathbf{z}|\boldsymbol{\phi}}\left[\|\hat{\boldsymbol{\theta}}\|_2^2\right] = \sum_{j=1}^{|\boldsymbol{\theta}|} \mathbb{P}[z_j \neq 0]\,\hat{\theta}_j^2$, where $\hat{\boldsymbol{\theta}}$ is a *gate-rescaled* version of $\boldsymbol{\theta}$ in order to "avoid extra shrinkage for the gates". Further discussion on the challenges of combining weight-decay and their proposed reparametrization can be found in Appendices H and I.

---

[§]Note that the $L_0$-norm of $\boldsymbol{\theta}$ is determined by that of $\mathbf{z}$. This is because for commonly used weight initialization and optimization schemes, $\tilde{\boldsymbol{\theta}} \neq 0$ almost surely.

# 3 Sparsity via $L_0$ Constraints

We favor formulating regularization goals as constraints, rather than as additive penalties with fixed scaling factors. We refer to these two approaches as *constrained* and *penalized*, respectively. Although a ubiquitous tool in machine learning, penalized formulations may come at the cost of hyper-parameter interpretability and are susceptible to intricate dynamics when incorporating multiple, potentially conflicting, sources of regularization.

## 3.1 Constrained Formulation

In contrast to the penalized objective of Louizos et al. [31] presented in Eq. (2), we propose to incorporate sparsity through constraints on the $L_0$-norm. We formulate an optimization problem that aims to minimize the model's expected empirical risk, subject to constraints on the expected $L_0$-norm of pre-determined parameter groups:

$$\min_{\tilde{\boldsymbol{\theta}},\boldsymbol{\phi}} \mathfrak{f}_{\text{obj}}(\tilde{\boldsymbol{\theta}},\boldsymbol{\phi}) \triangleq \mathbb{E}_{\mathbf{z}|\boldsymbol{\phi}}\left[\mathcal{L}_{\mathcal{D}}(\tilde{\boldsymbol{\theta}}\odot\mathbf{z})\right] \quad \text{s.t.} \quad \mathfrak{g}_{\text{const}}(\boldsymbol{\phi}_g) \triangleq \overbrace{\frac{\mathbb{E}_{\mathbf{z}_g|\boldsymbol{\phi}_g}\left[\|\mathbf{z}_g\|_0\right]}{\#(\tilde{\boldsymbol{\theta}}_g)}}^{L_0-\text{density}} \leq \epsilon_g \quad \text{for } g \in [1:G], \quad (3)$$

where $g$ denotes a subset of gates, $\#(\mathbf{x})$ counts the total number of entries in $\mathbf{x}$, and $\mathbf{x}_g$ denotes the entries of a vector $\mathbf{x}$ associated with the group $g$. See Appendix B for details on parameter grouping.

Note how the $\#(\tilde{\boldsymbol{\theta}}_g)$ factor in the constraint levels allows us to **interpret** $\epsilon_g$ as the maximum *proportion* of gates that are allowed to be active within group $g$, in expectation. We refer to $\epsilon_g$ as the ***target density*** of group $g$. Lowering the target density demands a sparser model and thus a (not necessarily strictly) more challenging optimization problem in terms of the best *feasible* empirical risk. Moreover, for any choice of $\epsilon_g \geq 0$, the feasible set in Eq. (3) is always non-empty; while values of $\epsilon_g \geq 1$ result in vacuous constraints.

We highlight one important difference between the constrained and penalized formulations. The penalized approach is *jointly* optimizing the training loss $\mathfrak{f}_{\text{obj}}(\tilde{\boldsymbol{\theta}},\boldsymbol{\phi})$ *and* the expected $L_0$-norm $\mathfrak{g}_{\text{const}}(\boldsymbol{\phi})$, due to their additive combination (mediated by $\lambda_{\text{pen}}$). Meanwhile, the constrained method focuses on obtaining the best possible model within a prescribed density level $\epsilon$: given two *feasible* solutions, the constrained formulation in Eq. (3) only discriminates based on the training loss. In other words, **we aim to *satisfy* the constraints, not to *optimize* them**.

## 3.2 Solving the Constrained Optimization Problem

We start by considering the (nonconvex-concave) Lagrangian associated with the constrained formulation in Eq. (3), along with the corresponding min-max game:

$$\tilde{\boldsymbol{\theta}}^*,\boldsymbol{\phi}^*,\boldsymbol{\lambda}_{\text{co}}^* \triangleq \underset{\tilde{\boldsymbol{\theta}},\boldsymbol{\phi}}{\operatorname{argmin}} \underset{\boldsymbol{\lambda}_{\text{co}}\geq 0}{\operatorname{argmax}} \; \mathfrak{L}(\tilde{\boldsymbol{\theta}},\boldsymbol{\phi},\boldsymbol{\lambda}_{\text{co}}) \triangleq \mathfrak{f}_{\text{obj}}(\tilde{\boldsymbol{\theta}},\boldsymbol{\phi}) + \sum_{g=1}^{G}\lambda_{\text{co}}^g\left(\mathfrak{g}_{\text{const}}(\boldsymbol{\phi}_g) - \epsilon_g\right), \quad (4)$$

where $\boldsymbol{\lambda}_{\text{co}} = [\lambda_{\text{co}}^g]_{g=1}^{G}$ are the (non-negative) Lagrange multipliers associated with each constraint.

A commonly used approach to optimize this Lagrangian is *simultaneous* gradient descent on $(\tilde{\boldsymbol{\theta}},\boldsymbol{\phi})$ and projected (to $\mathbb{R}^+$) gradient ascent on $\boldsymbol{\lambda}_{\text{co}}$ [30]:

$$[\tilde{\boldsymbol{\theta}}^{t+1},\boldsymbol{\phi}^{t+1}] \triangleq [\tilde{\boldsymbol{\theta}}^t,\boldsymbol{\phi}^t] - \eta_{\text{primal}}\nabla_{[\tilde{\boldsymbol{\theta}},\boldsymbol{\phi}]}\mathfrak{L}(\tilde{\boldsymbol{\theta}}^t,\boldsymbol{\phi}^t,\boldsymbol{\lambda}_{\text{co}}^t)$$

$$\hat{\boldsymbol{\lambda}}^{t+1} \triangleq \boldsymbol{\lambda}_{\text{co}}^t + \eta_{\text{dual}}\nabla_{\boldsymbol{\lambda}_{\text{co}}}\mathfrak{L}(\tilde{\boldsymbol{\theta}}^t,\boldsymbol{\phi}^t,\boldsymbol{\lambda}_{\text{co}}^t) = \boldsymbol{\lambda}_{\text{co}}^t + \eta_{\text{dual}}\left[\mathfrak{g}_{\text{const}}(\boldsymbol{\phi}_g^t) - \epsilon_g\right]_{g=1}^{G} \quad (5)$$

$$\boldsymbol{\lambda}_{\text{co}}^{t+1} \triangleq \max\left(0,\hat{\boldsymbol{\lambda}}^{t+1}\right)$$

The gradient update for $\boldsymbol{\lambda}_{\text{co}}$ matches the value of the violation of each constraint. When a constraint is satisfied, the gradient for its corresponding Lagrange multiplier is non-positive, leading to a reduction in the value of the multiplier.

**Negligible computational overhead.** Just as the penalized formulation of Louizos et al. [31], the update for $\tilde{\boldsymbol{\theta}}$ and $\boldsymbol{\phi}$ requires the gradient of the training loss and that of the expected $L_0$-norm. Hence, the cost of executing this update scheme is the same as the cost of a gradient descent update on the penalized formulation in Eq. (1), up to the negligible cost of updating the multipliers.

**Choice of optimizers.** We present simple gradient descent-ascent (GDA) updates in Eq. (5). However, our proposed framework is compatible with different choices for the primal and dual optimizers, including stochastic methods. Throughout our experiments, we opt for primal (model) optimizers which match standard choices for the different architectures. A choice of gradient ascent for the dual optimizer provided consistently robust optimization dynamics across all tasks. Detailed experimental configurations are provided in Appendix J. The evaluation and design of other optimizers, especially those for updating the Lagrange multipliers, is an interesting direction for future research.

**Oscillations.** The non-convexity of the optimization problem in Eq. (4) implies that a saddle point (pure strategy Nash Equilibrium) might not exist. In general, this can lead to oscillations and unstable optimization dynamics. Appendix F provides pointers to more sophisticated constrained optimization algorithms which achieve better convergence guarantees on nonconvex-concave problems than GDA. Fortunately, throughout our experiments we observed oscillatory behavior that quickly settled around feasible solutions. Empirical evidence of this claim is presented in Section 5.4.

**Extensibility.** Our proposed constrained formulation is "modular" in the sense that it is easy to induced other properties in the model's behavior beside sparsity (e.g. fairness [20, 5]) by prescribing them as *additional constraints*; much like extra additive terms in the penalized formulation. However, the improved interpretability and control afforded by the constrained approach removes the need to perform extensive tuning of the hyper-parameters to balance these potentially competing demands.

### 3.3 Dual Restarts

A drawback of gradient-based updates for optimizing the Lagrangian in Eq. (4) is that the constraint violations accumulate in the value of the Lagrange multipliers throughout the optimization, and continue to affect the optimization dynamics, *even after a constraint has been satisfied*. This results in an excessive regularization effect, which forces the primal parameters towards the *interior* of the feasible set. This behavior can be detrimental if we are concerned about minimizing the objective function and *satisfying* (but not minimizing!) the constraints.

To address this, we propose a ***dual restart scheme*** in which the Lagrange multiplier $\lambda_{\text{co}}^g$ associated with a constraint $\mathfrak{g}_{\text{const}}(\phi_g) \leq \epsilon_g$ is set to 0 whenever the constraint is satisfied; rather than waiting for the "negative" gradient updates ($\mathfrak{g}_{\text{const}}(\phi_g) - \epsilon_g < 0$ when feasible) to reduce its value. Formally,

$$\left[\lambda_{\text{co}}^{t+1}\right]_g \triangleq \begin{cases} \max\left(0, \left[\lambda_{\text{co}}^t\right]_g + \eta_{\text{dual}}\left(\mathfrak{g}_{\text{const}}(\phi_g^t) - \epsilon_g\right)\right), & \text{if} \quad \mathfrak{g}_{\text{const}}(\phi_g^t) > \epsilon_g \\ 0, & \text{otherwise} \end{cases} \tag{6}$$

Dual restarts remove the contribution of the expected $L_0$-norm to the Lagrangian for groups $g$ whose constraints are satisfied, so that the optimization may focus on improving the predictive performance of the model. In fact, this dual restart strategy can be theoretically characterized as a *best response* (in the game-theoretic sense) by the dual player. The effect of dual restarts in the optimization dynamics is illustrated in Section 5.3 and Appendix G.

## 4 Related Work

**Min-max optimization.** Commonly used methods for solving constrained convex optimization problems [1, 11, 21] make assumptions on the properties of the objective function, constraints or feasible set. In this work, we focus on applications involving neural networks, leading to the violation of such assumptions. We rely on a GDA-like updates for optimizing the associated Lagrangian. However, our proposed formulation can be readily integrated with more sophisticated/theoretically supported algorithms for constrained optimization of non-convex-concave objectives, such as the extragradient method [24]. Further discussion on guarantees and alternative algorithms for min-max optimization is provided in Appendix F.

**Model sparsity.** Learning sparse models is a rich research area in machine learning. There exist many different approaches for obtaining sparse models. Magnitude-based methods [41, 18] perform one or more rounds of pruning, by removing the parameters with the lowest magnitudes. Popular non-magnitude based techniques include [26, 17, 34]. Structured pruning methods [6, 37, 31, 29], remove entire neurons/channels rather than *individual* parameters. More recently, the Lottery Ticket Hypothesis [12] has sparked interest in techniques that provide the storage and computation benefits of sparse models *directly during training* [35, 8]. However, finding "good" sparse sub-networks at initialization remains a central challenge for these techniques [13, 33].

**Controllable sparsity.** Magnitude pruning [18, 29] can achieve arbitrary levels of sparsity "by design" since it removes *exactly* the proportion of parameters with lowest magnitudes in order to match the desired density. However, the magnitude pruning method experiences certain shortcomings: ① retaining performance usually involves several round of fine-tuning[¶]; ② it relies on the *assumption* that magnitude (of filters or activations) is a reasonable surrogate for parameter importance; and ③ it lacks the "extensibility" property of our constrained formulation: it is not immediately evident how to induce other desired properties in the model, besides sparsity.

Note that several extensions of the basic magnitude pruning method have been proposed. Zhu and Gupta [47] start from a partially or fully pre-trained model and consider a sparsification scheme in which the network density is gradually reduced, while fine-tuning the model to compensate for any potential loss in performance due to pruning. Wang et al. [44] start by *identifying* the parameters to be removed by applying magnitude pruning on a pre-trained model. However, rather than pruning the model immediately, the authors propose to fine-tune the model with an adaptive $L_2$-penalty. The weight of this penalty is increased over time for the previously identified parameters, leading their magnitudes to decrease during the fine-tuning process.

**Sparsity via constrained optimization.** Previous works have cast the task of learning sparse models as the solution of a constrained optimization problem. Carreira-Perpinan and Idelbayev [3] consider a reformulation of the constrained optimization problem using "auxiliary variables", and assume that the constraints enjoy an efficient proximal operator. Their empirical evaluation is limited to low-scale models and datasets.

Zhou et al. [46] adopt a constrained formulation similar to ours, although based on a different reparametrization of the gates. The authors tackle the constrained problem via projected gradient descent by cleverly exploiting the existence of an efficient projection of the gate parameters onto their feasible set. However, the applicability of their method is limited to constraints with an efficiently-computable projection operator.

Lemaire et al. [27] consider "budget-aware regularization" and tackle the constrained problem using a barrier method. Although originally inspired by a constrained approach, their resulting training objective corresponds to a penalized method with a penalty factor that requires tuning, in addition to the choice of barrier function.

**Other constrained formulations in ML.** Constrained formulations can be used to prescribe desired behaviors or properties in machine learning models. Nandwani et al. [36] study the problem of training deep models under constraints on the network's predicted labels, and approach the constrained problem in practice through a min-max Lagrangian formulation. Incorporating these constraints during training allows them to inject domain-specific knowledge into their models across several tasks in natural language processing.

Fioretto et al. [10] consider a wide range of applications spanning from optimal power flow in energy grids, to the training of fair classification models. Their work demonstrates how Lagrangian-based methods can be complementary to deep learning by effectively enforcing complex physical and engineering constraints.

Cotter et al. [5] train models under constraints on the prediction rates of the model over different datasets. Note that the sparsity constraints we study in this paper depends only on properties of the *parameters* and not on the *predictions* of the model. We would like to highlight that the notion of *proxy constraints* introduced by Cotter et al. [5] can enable training models based on constraints on their actual test time density, rather than the surrogate expected $L_0$-norm metric.

## 5 Experiments

The main goal of our work is to train models that attain good predictive performance, while having a fine-grained command on the sparsity of the resulting model. In this section we present a comparison with the work of Louizos et al. [31][‖]; we explore the stability and controllability properties of our Lagrangian-based constrained approach, along with the effect of our proposed dual restarts heuristic. Finally, we present empirical evidence which demonstrates that our method successfully retains its interpretability and controllability advantages when applied to large-scale models and datasets.

---

[¶]This re-training overhead makes magnitude pruning less appealing compared with in-training alternatives, since magnitude pruning is typically performed given an *already fully trained* model.

[‖]See a comparison to other sparsity methods therein, along with the survey of Gale et al. [14].

## 5.1 Experimental Setup

**Experiment configuration and hyperparameters.** Details on our implementation, hyperparameter choices and information on the network architectures can be found in Appendices A, B, C, D and J.

**Model- and layer-wise settings.** We present experiments using two kinds of constraints: one *global* constraint on the proportion of active gates throughout the entire model; or several *local* constraints prescribing a maximum density at each layer. Note that for models such as ResNet50, the layer-wise setting involves handling 48 constraints. The experiments below demonstrate that **our constrained approach can gracefully handle from a single constraint up to dozens of constraints** in a unified way and still achieve controllable sparsity for *each* of the layers/model. This level of control is an intractable goal for penalized methods: as demonstrated in Appendix E, even trying to tame *one* constraint via a penalty factor can be prohibitively expensive.

$L_0$**-regularization for residual models.** ResNets have been a challenging setting for $L_0$-penalty based methods. Gale et al. [14] trained WideResNets [45] and ResNet50 [19] using the penalized $L_0$-regularization framework of Louizos et al. [31], and reported being unable to produce sparse ResNets without significantly degrading performance.

We propose two simple adjustments that enable us to successfully train WRNs and ResNets with controllable sparsity, while retaining competitive performance: ① increasing the learning rate of the stochastic gates; and ② removing the gradient contribution of the weight decay penalty towards the gates. Appendices H and I provide detailed analysis and empirical validation of these two modifications. We integrate these adjustments in all experiments involving residual models below.

**Obtaining test-time models.** Appendix D describes our procedure to transform a model with stochastic gates into a deterministic, test-time model. The measurements of retained parameters and MACs (multiply-accumulate operations) percentages reported in the tables and figures below, are computed for the deterministic, purged, test-time models.

## 5.2 Proof of Concept Experiments on MNIST

We begin by comparing the behavior of our method with that of Louizos et al. [31] in the simple setting of training MLP and LeNet5 architectures on the MNIST dataset. The authors report the size of their pruned architectures found using the penalized formulation. In this section we aim to showcase the *controllability* advantages of our constrained approach. We manually computed the corresponding model-wise or layer-wise density levels achieved by the reported architectures of Louizos et al. [31] and used these values as the target density levels for our constrained formulation.

Table 1: Achieved density levels and performance for sparse MLP and LeNet5 models trained on MNIST for 200 epochs. Metrics aggregated over 5 runs. [†]Results by Louizos et al. [31] with $N$ representing the training set size (see Appendix C).

| Architecture | Grouping | Method | Hyper-parameters | Pruned architecture | Val. Error (%) best | at 200 epochs (avg $\pm$ 95% CI) |
|---|---|---|---|---|---|---|
| MLP 784-300-100 | Model | Pen. | [†]$\lambda_{pen} = 0.1/N$ | 219-214-100 | 1.4 | – |
| | | Const. | $\epsilon = 33\%$ | 198-233-100 | 1.36 | $1.77 \pm 0.08$ |
| | Layer | Pen. | [†]$\lambda_{pen} = [0.1, 0.1, 0.1]/N$ | 266-88-33 | 1.8 | – |
| | | Const. | $\epsilon = [30\%, 30\%, 30\%]$ | 243-89-29 | 1.58 | $2.19 \pm 0.12$ |
| LeNet5 20-50-800-500 | Model | Pen. | [†]$\lambda_{pen} = 0.1/N$ | 20-25-45-462 | 0.9 | – |
| | | Const. | $\epsilon = 10\%$ | 20-21-34-407 | 0.56 | $1.01 \pm 0.05$ |
| | Layer | Pen. | [†]$\lambda_{pen} = [10, 0.5, 0.1, 0.1]/N$ | 9-18-65-25 | 1.0 | – |
| | | Const. | $\epsilon = [50\%, 30\%, 70\%, 10\%]$ | 10-14-224-29 | 0.7 | $0.91 \pm 0.05$ |

Table 1 displays the results of our constrained method and the reported metrics for the penalized approach. Note that, as desired, the pruned models obtained using the constrained formulation resemble closely the "target architecture sizes" reported by Louizos et al. [31]. Moreover, our method does not cause any loss in performance with respect to the penalized approach. This final observation will be confirmed for larger-scale tasks in later sections.

Note that the goal of this section is to demonstrate that our constrained approach can achieve arbitrary sparsity targets "in one shot" (i.e. without trial-and-error tuning) and without inducing any

compromise in the predictive performance of the resulting models. Comprehensive experiments for MLP and LeNet5 models on MNIST across a wide range of sparsity levels for model- and layer-wise constraints are presented in Appendix K.1.

## 5.3 Training Dynamics and Dual Restarts

We now discuss the effect of the dual restarts scheme introduced in Section 3.3 on the training dynamics of our constrained formulation. Fig. 2 illustrates the training of a convolutional network on MNIST under a $30\%$ model-wise density constraint when using (blue) or not (orange) dual restarts.

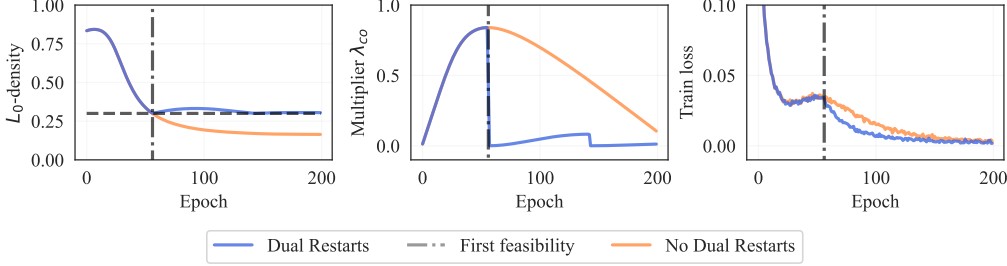

Figure 2: Effect of dual restarts for training a LeNet5 on MNIST with a model-wise target density $\epsilon_g = 30\%$ (horizontal dashed line). The accumulation of the constraint violations in the Lagrange multiplier leads to excessive sparsification when the model satisfies the constraint. Restarting the Lagrange multiplier allows the model to concentrate on improving the training loss.

We initialize the Lagrange multipliers to zero. Therefore, at the beginning of optimization there is no contribution from the $L_0$-norm in the Lagrangian (see Eq. (4)), and the optimization focuses on improving the training loss. As the optimization progresses, the constraint violations are accumulated in the value of the Lagrange multiplier. When the Lagrange multiplier is sufficiently large, the importance of satisfying the constraints outweighs that of optimizing the training loss.[**] In consequence, the model density decreases. As the model reaches the desired sparsity level, $\lambda_{co}$ stops increasing.

Up until the time at which the model is first feasible, the multiplier value accumulates the constraint violations (scaled by the dual learning rate). Once the model is feasible, the constraint violation $\mathfrak{g}_{const}(\phi_g) - \epsilon < 0$ becomes negative, leading to a *decrease* in the Lagrange multiplier. However, at this stage, the Lagrange multiplier is large due to the accumulated constraint violations. This confers a higher relative importance to the gradient of the constraints over that of the training loss: the larger multiplier encourages to reduce the *constraints even if they are already being satisfied*.

Our proposed dual restart heuristic reduces the Lagrange multiplier to zero whenever the constrained is satisfied, allowing the training to focus on minimizing the training loss faster. Although this heuristic may lead to slightly unfeasible solutions, as demonstrated throughout our experiments, our models remain consistently below (or close to) the required $L_0$-density levels.

## 5.4 Stable Constraint Dynamics

Despite the theoretical risk of oscillatory dynamics commonly associated with iterative constrained optimization methods, we consistently observed quickly stabilizing behavior in our experiments. Fig. 3 shows the density levels throughout training for a layer of a WideResNet-28-10 trained on CIFAR-10 (right), and the model-wise density of a ResNet18 trained on TinyImageNet (left).

The desired density levels are successfully achieved over a wide range of targets, and the constraint dynamics stabilize quickly. These dynamics were consistent across all our architectures and datasets.

---

[**]Note that the Lagrange multiplier influences the update of the model parameters by *dynamically adjusting* the relative importance of the gradient of the training loss with respect to the gradient of the constraint. In the penalized method this relative importance is fixed.

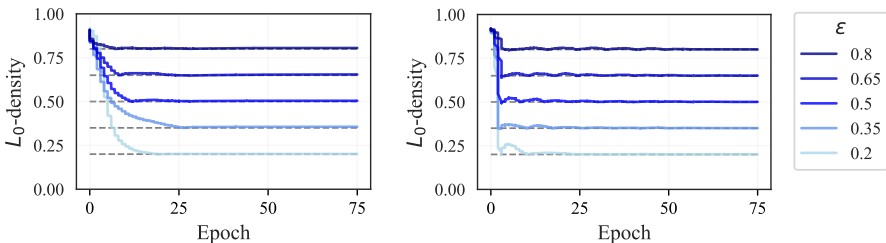

Figure 3: Density levels for a ResNet18 model (left; trained with a model-wise constraint) and the last sparsifiable layer of a WideResNet-28-10 model (right; trained with layer-wise constraints).

## 5.5 Large-scale Experiments

We now demonstrate the scalability of our method to more challenging settings: we consider (Wide)ResNet models on the CIFAR-{10, 100}, TinyImageNet [28] and ImageNet [7] datasets. Comprehensive experiment are provided in Appendix K.

**CIFAR-{10, 100} and TinyImageNet.** Figures 1 and 4 display the results for a ResNet18 model trained on Imagenet, and a WideResNet-28-10 trained on CIFAR-10, respectively. The left column shows the alignment between the achieved and desired densities (as expected proportion of *active gates* in the model). Our method (in blue) provides a robust control over the range of densities. In contrast, the penalized method (in red) exhibits an unreliable dependency between the penalty coefficient and the achieved density: when increasing the coefficient $\lambda_{\text{pen}}$, the achieved density seems to be insensitive to $\lambda_{\text{pen}}$ for several orders of magnitude until it starts considerably changing. This brittle sensitivity profile limits the potential of the penalized method for controlling sparsity.

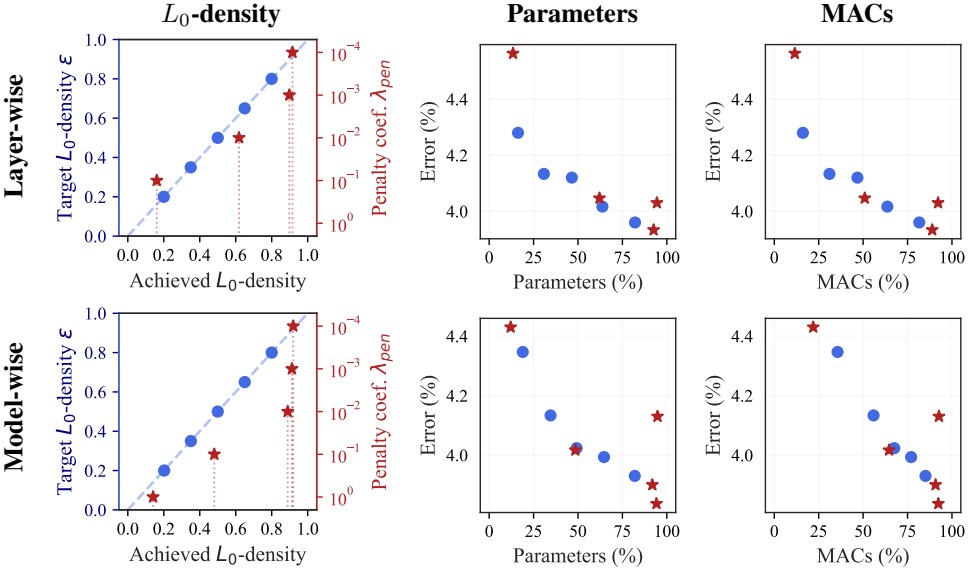

Figure 4: Training sparse WideResNet-28-10 models on CIFAR-10.

Columns two and three display the number of parameters and MACs (multiply-accumulate operations) of the resulting purged models, as a proportion of those of the fully-dense baseline model. Note that, while retaining a similar proportion of parameters, layer-wise constraints lead to a larger reduction in the number of MACs, compared to the model-wise case. This is because layer-wise constraints induce a strict, homogeneous sparsification of all the modules of the network; while the model-wise setting can allow for a more flexible allocation of the parameter budget across different layers.

**ImageNet.** We conducted experiments on ImageNet [7] with a ResNet50 architecture. We compare with layer-wise structured magnitude pruning [29][††]. The results are presented in Table 2.

---

[††]For *each layer*, we remove the filters with the $(1 - \epsilon)$ lowest $L_1$-norms to achieved the desired $\epsilon$ density.

Note that experiments with layer-wise constraints correspond to optimization problems with 48 constraints (one for each sparsifiable layer). We highlight the number of constraints since tuning such a large number of penalty coefficients is an intractable challenge when using the penalized method.

Table 2: ResNet50 models on ImageNet with structured sparsity. "Fine-tuning" for zero epochs means *no* fine-tuning.

| Target Density | Method | $L_0$-density (%) | Params (%) | MACs (%) | Best Val. Error (%) After fine-tuning for # epochs | | | |
|---|---|---|---|---|---|---|---|---|
| | | | | | 0 | 1 | 10 | 20 |
| – | Pre-trained Baseline | 100 | [25.5M] | [$4.12 \cdot 10^9$] | 23.90 | | | – |
| $\epsilon = 90\%$ | Const. *Model-wise* | 90.36 | 88.06 | 91.62 | **24.68** | | | – |
| | Const. *Layer-wise* | 90.58 | 87.07 | 85.97 | 24.97 | | | – |
| | L1-MP *Layer-wise* | – | 85.94 | 84.99 | 38.74 | 25.38 | 24.69 | **24.68** |
| $\epsilon = 70\%$ | Const. *Model-wise* | 70.78 | 64.41 | 76.50 | **25.53** | | | – |
| | Const. *Layer-wise* | 70.36 | 61.91 | 58.59 | 26.98 | | | – |
| | L1-MP *Layer-wise* | – | 62.15 | 59.85 | 97.78 | 29.04 | 26.80 | 26.14 |
| $\epsilon = 50\%$ | Const. *Model-wise* | 50.18 | 42.47 | 58.00 | **27.51** | | | – |
| | Const. *Layer-wise* | 50.70 | 43.15 | 38.25 | 27.89 | | | – |
| | L1-MP *Layer-wise* | – | 43.47 | 39.76 | 99.75 | 36.21 | 29.98 | 29.16 |
| $\epsilon = 30\%$ | Const. *Model-wise* | 30.31 | 31.81 | 42.05 | **29.65** | | | – |
| | Const. *Layer-wise* | 31.44 | 30.16 | 23.74 | 31.71 | | | – |
| | L1-MP *Layer-wise* | – | 29.86 | 24.80 | 99.89 | 56.11 | 36.90 | 34.74 |

Just like the magnitude pruning method, our proposed approach successfully delivers the desired levels of sparsity in this challenging task. To the best of our knowledge, our work constitutes the first instance of successfully learning ResNet50 models using the $L_0$ reparametrization of Louizos et al. [31] for structured sparsity while retaining high accuracy.

Our results clearly demonstrate that the constrained $L_0$ formulations can obtain large levels of structured parameter reduction while preserving performance. Table 2 shows a quick degradation in performance for the magnitude pruning method, and highlights the need for fine-tuning in heuristic-based pruning techniques.

## 5.6 Unstructured Sparsity

Appendix L contains experiments with unstructured sparsity (i.e. one gate per parameter, rather than per neuron/activation map) for the MNIST and TinyImageNet datasets. These experiments show that the controllability advantages of our constrained formulation apply in the unstructured regime. Recall that Gale et al. [14] report an apparent dichotomy between sparsity and performance when training (residual) models with unstructured sparsity using the $L_0$ reparametrization of Louizos et al. [31]. Our experimental results demonstrate that it is in fact possible to achieve high levels of sparsity *and* predictive performance.

## 6 Conclusion

We resort to a constrained optimization approach as a tool to overcome the controllability shortcomings faced by penalty-based sparsity methods. Along with a reliable control of the model density, this technique provides a more interpretable hyper-parameter and removes the need for expensive iterative tuning. We adopt the $L_0$ reparametrization framework of Louizos et al. [31] and integrate simple adjustments to remedy their challenges at training (Wide)ResNet models. Our proposed method succeeds at achieving the desired sparsity with no compromise on the model's performance for a broad range of architectures and datasets. These observations position the constrained approach as a solid, practical alternative to popular penalty-based methods in modern machine learning tasks.

## Acknowledgments and Disclosure of Funding

This work was partially supported by the Canada CIFAR AI Chair Program and by an IVADO PhD Excellence Scholarship. Simon Lacoste-Julien is a CIFAR Associate Fellow and Yoshua Bengio is a Program Co-Director in the Learning in Machines & Brains program.

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
