# Appendix

# A   Reparametrization of the Gates

Louizos et al. [31] introduce the hard concrete distribution for modeling the stochastic gates $\mathbf{z}$. Consider a concrete random variable $s_j \sim q(\cdot \mid (\phi_j, \beta))$, given a fixed $0 < \beta < 1$. This variable is then stretched and clamped, resulting in a mixed distribution with point masses at $0$ and $1$, and a continuous density over $(0, 1)$.

Formally, given $U_j \sim \text{Unif}(0, 1)$ and hyper-parameters $\gamma < 0 < 1 < \zeta$,

$$s_j = \text{Sigmoid}\left( \frac{1}{\beta} \log \left( \frac{\phi_j U_j}{1 - U_j} \right) \right); \quad \mathbf{z} = \text{clamp}_{[0,1]}(\mathbf{s}(\zeta - \gamma) + \gamma)) \tag{7}$$

The stochastic nature of $\mathbf{z}$ entails a model which is itself *stochastic*. Therefore, both its $L_0$-norm and predictions are random quantities. Nonetheless, the specific choice of re-parameterization in Eq. (7) allows for the training loss in Eq. (2) to be estimated using Monte Carlo samples.

Moreover, Louizos et al. [31] show that the expected $L_0$-norm can be expressed in closed-form as:

$$\mathbb{E}_{\mathbf{z}|\boldsymbol{\phi}}\left[ \|\boldsymbol{\theta}\|_0 \right] = \sum_{j=1}^{|\boldsymbol{\theta}|} \mathbb{P}[z_j \neq 0] = \sum_{j=1}^{|\boldsymbol{\theta}|} \text{Sigmoid}\left( \log \phi_j - \beta \log \frac{-\gamma}{\zeta} \right) \tag{8}$$

The probability distribution of the gates has both learnable and fixed parameters. Table 3 specifies the values of the fixed parameters employed throughout this work, following Louizos et al. [31].

| Parameter | $\gamma$ | $\zeta$ | $\beta$ |
|---|---|---|---|
| Value | -0.1 | 1.1 | 2/3 |

Table 3: Fixed parameters of the hard concrete distribution.

## A.1   Choice of Gates at Test-Time

Recall that the stochastic reparametrization induces a distribution over models. We suggest a natural way of "freezing" the network gates so as to obtain a *deterministic* predictor when evaluating the model on unseen datapoints: replace each stochastic gate by its median.

$$\hat{z}_j = \min \left( 1, \, \max \left( 0, \, \text{Sigmoid}\left( \frac{\log(\phi_j)}{\beta} \right) (\zeta - \gamma) + \gamma \right) \right) \tag{9}$$

Note that these medians may be *fractional*, i.e., $z_j \in [0, 1]$. Nonetheless, as shown in Appendix I, for trained sparse models we observe the medians to be highly concentrated at $0$ and $1$.

Louizos et al. [31, Eq. (13)] originally proposed a similar approach for obtaining a deterministic test-time model (without an explicit motivation for their choice). Their proposal differs from ours in that they do not perform a division by $\beta$, and thus their test-time gates are not the median (nor the mean) of the gate distributions.

In our preliminary experiments the division by $\beta$ (under the settings of Table 3) did not induce significant changes in behavior. However, we opt for Eq. (9) in our experiments based on its concise theoretical motivation.

## A.2   Initialization of the Gates

Louizos et al. [31] introduce a hyper-parameter $\rho_{\text{init}} \in (0, 1)$ which determines the initialization of the parameter $\phi_j$ of the hard concrete distribution of the gates (see Appendix A). Concretely, the gate parameters $\phi_j$ are initialized as:

$$\log \phi_j = \log \left( \frac{1 - \rho_{\text{init}}}{\rho_{\text{init}}} \right) + \mathcal{N}(0, 10^{-2}) \tag{10}$$

Note that in practice, the optimization variable is $\log \phi_j$ (rather than $\phi_j$) as this sidesteps having to preserve the non-negativity of $\phi_j$.

The choice of $\rho_{\text{init}}$ has an inverse relationship with the probability of a gate being active at initialization. For simplicity, we ignore the small additive noise in the initialization of $\log \phi_j$, and let $\psi = (-\gamma/\zeta)^\beta$. Formally, the influence of the hyper-parameter $\rho_{\text{init}}$ at initialization is given by:

$$\mathbb{P}[z_j \neq 0] = \text{Sigmoid}\left(\log\left(\frac{1 - \rho_{\text{init}}}{\rho_{\text{init}}}\right) - \log\left(\frac{-\gamma}{\zeta}\right)^\beta\right) = \frac{1 - \rho_{\text{init}}}{1 - (1 - \psi)\rho_{\text{init}}}. \qquad (11)$$

## B  Schemes for Grouping Parameters

We consider two schemes for grouping gates: a) with a single constraint/penalty on the proportion of active gates of the model, or b) with separate constraints/penalties for each layer. These groupings are referred to as *model-wise* and *layer-wise*.

For the penalized method, the corresponding optimization problems are given by:

**Model-wise grouping**

$$\min_{\tilde{\theta}, \phi} \quad \mathfrak{f}_{\text{obj}}(\tilde{\theta}, \phi) + \lambda_{\text{pen}} \, \mathfrak{g}_{\text{const}}(\phi)$$

**Layer-wise grouping**

$$\min_{\tilde{\theta}, \phi} \quad \mathfrak{f}_{\text{obj}}(\tilde{\theta}, \phi) + \sum_{g=1}^{\texttt{num\_layers}} \lambda_{\text{pen}}^g \, \mathfrak{g}_{\text{const}}(\phi_g)$$

For the constrained method, the corresponding optimization problems are given by:

**Model-wise grouping**

$$\min_{\tilde{\theta}, \phi} \quad \mathfrak{f}_{\text{obj}}(\tilde{\theta}, \phi)$$
$$\text{s.t.} \quad \mathfrak{g}_{\text{const}}(\phi) \leq \epsilon$$

**Layer-wise grouping**

$$\min_{\tilde{\theta}, \phi} \quad \mathfrak{f}_{\text{obj}}(\tilde{\theta}, \phi)$$
$$\text{s.t.} \quad \mathfrak{g}_{\text{const}}(\phi_g) \leq \epsilon_g \quad \text{for } g \in [1 : \texttt{num\_layers}]$$

For example, consider a $[d_{\text{in}}, d_{\text{hid}}, d_{\text{out}}]$ 1-hidden layer MLP with input-neuron sparsity on both of its two fully connected layers. For simplicity, we ignore the bias in the description below.

- **Grouping at the layer level** (akin to "$\lambda$ sep." in [31]), would yield $G = 2$ groups, with $d_{\text{in}}$ gates in group $g = 1$. Each gate in group 1 is shared across $d_{\text{hid}}$ parameters in $\tilde{\theta}$, thus $\#(\tilde{\theta}_1) = d_{\text{in}} \cdot d_{\text{hid}}$. Similarly for $g = 2$.
- **Grouping at the model level** (akin to "$\lambda \propto \frac{1}{N}$" in [31]). corresponds to the case of $G = 1$ group comprising with $d_{\text{in}} + d_{\text{hid}}$ gates. Finally, $\#(\tilde{\theta}_1) = d_{\text{in}} \cdot d_{\text{hid}} + d_{\text{hid}} \cdot d_{\text{out}}$ gives the total number of parameters in the network.

A similar analysis holds for the case of output feature-map sparsity used in convolutional layers.

## C  Normalizing the $L_0$-norm

Louizos et al. [31] normalize the expected $L_0$-norm of model parameters with respect to the *training set size $N$*, and not with respect to the *number of parameters*. This is done by selecting a $\lambda_{\text{pen}} = \lambda/N$. In contrast, as stated in Eq. (3), we favor normalizing by the total number of parameters in the model $\#(\tilde{\theta})$. This yields an expected $L_0$-density consistently in the range $[0, 1]$ regardless of model architecture.

Optimization problems corresponding to each of these normalization schemes (grouping gates model-wise, for illustration) are given by:

**Ours (Penalized)**

$$\min_{\tilde{\theta}, \phi} \quad \mathfrak{f}_{\text{obj}}(\tilde{\theta}, \phi) + \lambda_{\text{pen}} \frac{\mathbb{E}_{\mathbf{z} \mid \phi}[\|\mathbf{z}\|_0]}{\#(\tilde{\theta})}$$

**Louizos et al. [31]**

$$\min_{\tilde{\theta}, \phi} \quad \mathfrak{f}_{\text{obj}}(\tilde{\theta}, \phi) + \lambda_{\text{pen}} \frac{\mathbb{E}_{\mathbf{z} \mid \phi}[\|\mathbf{z}\|_0]}{N}$$

Therefore, a $\lambda_{\text{pen}} = \lambda$ in the context of our work does *not* correspond to the same regularization factor as choosing $\lambda_{\text{pen}} = \lambda/N$ in Louizos et al. [31]. For details on the number of parameters of each architecture and the associated number of training examples, see Appendix J.1. Note that in certain settings (e.g. CIFAR-10/100) the number of training points and number of parameters of the model can differ by several orders of magnitude.

# D Purging Models

In this section, we describe how we transform a model with stochastic gates $\mathbf{z}$ and free parameters $\tilde{\boldsymbol{\theta}}$ into a deterministic test-time model. For conciseness, we present the procedure for convolutional layers. The case of fully connected layers is analogous: simply consider the parameter groups to be "all those weights connecting an input neuron to any neuron in the next layer".

The procedure for removing filters from the $i$-th convolutional layer is as follows:

1. For each filter in the layer, compute the test-time value of its associated gate as described in Appendix A.1.

2. Gates with medians $\hat{z}_j = 0$ are considered inactive. The value of an active gate $\hat{z}_j > 0$ is "absorbed" multiplicatively by its associated weights.

3. Prune the filters associated with inactive gates and their corresponding activation maps. The kernel entries in the next convolutional layer associated with the pruned channels in layer $i$ are also removed. If present, weights of the adjacent batch normalization layer are removed.

4. A new kernel matrix is created for both the $i$-th and $(i + 1)$-th layer, and the remaining kernel weights are copied to the new model.

The pruning procedure described above guarantees equivalent outputs for the network before and after pruning under the assumption that the element-wise activation function $h$ used in the network satisfies $h(0) = 0$, as is the case for ReLU activations.

**Double sparsification.** We highlight that the pruning of filters of the subsequent layer in step 3 happens *regardless of whether the following layer is sparsifiable* or not. This observation implies that if a model contains adjacent sparsifiable convolutional layers, the resulting number of active parameters in the second layer will be affected by its output sparsity rate, *as well as the output sparsity rate of the first layer*.

This "double sparsification" leads to a subtle issue when studying the relationship between the proportion of active gates in the model and the *effective* number of active parameters. For example, if $80\%$ of the (output) gates of layer $i$ are active, and $70\%$ of the (output) gates of layer $i + 1$ are active, the effective number of active parameters in layer $i + 1$ will be $\sim 56\% = (0.8 \cdot 0.7)$ and not $70\%$!

Note that this procedure is identical to that of Li et al. [29, §3.1]. However, our selection of filters to remove is based a sparsity pattern learned during training, rather than motivated by a heuristic ranking of the filter norms. We use a similar language and presentation to facilitate the comparison.

The double sparsification effect arises due to performing structured pruning in adjacent layers, while considering groupings (i.e. one gate per output channel) which disregard the sparsity from the previous and next layers. We discuss this issue to provide clarity when analyzing our results regarding controllability: the goal of our constrained formulation is to achieve (at most) a certain proportion of expected active *gates*. Our experiments demonstrate that we can indeed achieve said control (compare target density $\epsilon$ with $L_0$-density columns throughout all tables). Addressing the issue of "double sparsification" in structured pruning is beyond the scope of our work.

**Parameters and MACs.** To achieve fair comparisons between models trained using different techniques (e.g. penalized, constrained or magnitude pruning) we apply the same purging procedure to all models. For all of the experiments presented in the paper, the reported parameter and MAC[‡‡] counts are calculated based on the deterministic, purged models. The number of MACs corresponds to the number of multiply-accumulate operations involved in a *forward* pass through the network.

In the case of magnitude pruning, steps 1 is replaced with the ranking-and-thresholding operation of Li et al. [29]. This results (conceptually) in binary 0-1 values for the gates associated with each of the filters, required in step 2.

---

[‡‡]A MAC operation modifies an accumulator $a$ as $a \leftarrow a + (b \times c)$.

# E  Bisection Search

We execute a bisection search algorithm on the (logarithmic) value of $\lambda_{\text{pen}}$, aiming to achieve a model $L_0$-density of $50\%(\pm 1\%)$. Fig. 5 shows the results for experiments with parameter groupings model- and layer-wise.

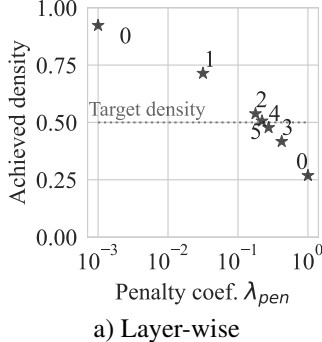
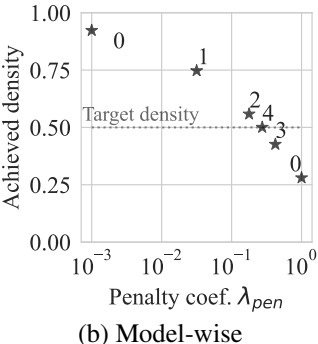

a) Layer-wise  (b) Model-wise

Figure 5: Iterations of bisection search on the logarithmic $\lambda_{\text{pen}}$ space for achieving a model density of $50\%(\pm 1\%)$. Annotations represent iteration indices, with endpoints labelled as 0. We report the $L_0$-density of MLPs after 150 training epochs. On the left, parameters are grouped layer-wise and groups share a fixed $\lambda_{\text{pen}}$; on the right, parameters are grouped at the model level.

Although bisection search successfully finds a penalty value which achieves the desired density, it required the execution of at least 6 complete training cycles to be within $1\%$ of the target. Performing such a high number of repeated experiments for tuning $\lambda_{\text{pen}}$ can be in-admissibly costly in real applications. Moreover, we chose the endpoint values such that their resulting densities enclosed the target, reducing the difficulty of the search problem. While bisection search is by no means the optimal approach to adjust $\lambda_{\text{pen}}$, these experiments highlight the tunability challenges associated with penalized methods.

# F  Constrained Optimization: Theory and Further Algorithms

Recall that a pure Nash equilibrium of the min-max game in Eq. (4) corresponds to a saddle point of the Lagrangian and determines an optimal, feasible solution [43]. However, for non-convex problems such pure Nash equilibria might not exist, and thus simultaneous gradient descent-ascent (GDA) updates can *potentially* lead to oscillations in the parameters. [5, 30].

There has been extensive research in the saddle-point optimization community studying in these type of problems. In particular, for convex-concave problems there are (non-)asymptotic convergence guarantees for averaged iterates from GDA with equal step-sizes [24, 4, 38]. Lin et al. [30] present a comprehensive bibliography of studies focusing on nonconvex-concave problems, as is the case for the Lagrangian in Eq. (4). The authors also present non-asymptotic complexity results showing that two-timescale GDA can find stationary points for nonconvex-concave minimax problems efficiently.

Cotter et al. [5] propose an algorithm for non-convex constrained problems that returns an approximately optimal and feasible solution, consisting of a pair of mixed strategies (with support size of at most $m + 1$ for a problem with $m$ constraints). Experimentally we observed convergent, non-oscillatory behavior when using simultaneous updates for solving the problem in Eq. (4) employing *pure strategies*, i.e., a single instance of primal and dual variables. This is discussed in detail in Section 5.4 and Appendix G.

Other approaches, such as extra-gradient [24], provide better convergence guarantees for games like Eq. (4), compared to GDA. However, extra-gradient requires twice as many gradient computations per parameter update and the storage of an auxiliary copy of all trainable parameters. Nonetheless, extrapolation from the past [16] enjoys similar convergence properties to extra-gradient without requiring a second gradient computation. Our preliminary experiments showed no significant difference in performance when using extra-gradient-based updates. These techniques can be useful for mitigating oscillatory behavior when applying Lagrangian-based optimization to other constrained problems.

# G Training Dynamics and Dual Restarts

In this section we provide further details on the training dynamics of our gradient descent-ascent approach for solving the Lagrangian in Eq. (4). Fig. 6 displays the training dynamics for a LeNet model on the MNIST dataset using model-wise density constraints of 30% and 70%, as well as whether or not using dual restarts. The experimental setup for this section matches that of Appendix J.3.

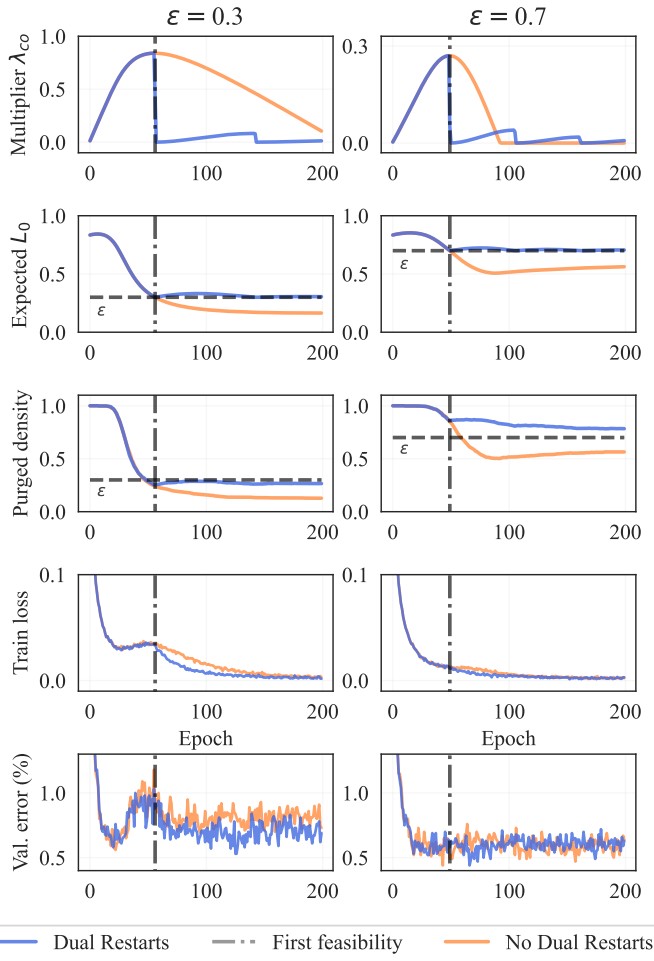

Figure 6: Effect of the dual restarts scheme on the training dynamics for LeNet models on MNIST using model-wise constraints with target densities of 30% and 70%.

We employ the same learning rate for both cases. Since the left column corresponds to a constraint yielding a more sparse model, the initial constraint violations are larger. In consequence, the magnitude of the Lagrange multiplier, which accumulates the constraint violations, is also larger for the 30% case (compare the scale of the vertical axis in the plots of the second row).

The horizontal dashed line in the first row signals the desired density for each of the cases. Note that all models become feasible: blue and orange lines are at or below the density target.

However, *not* employing dual restarts leads to the model being "excessively" sparsified: the orange line overshoots past the desired density level. While in principle this behavior is not "wrong" since the constraint is satisfied, it can lead to slow learning and decreased performance. Note that our constrained formulation leads to a natural monotonicity property in the constraints: if $\epsilon' < \epsilon$, the best performance achievable by a $\epsilon$-dense model is greater than or equal to the best performance achievable by an $\epsilon'$-dense model.

When dual restarts are applied, the contribution of the accumulated constraint violation to the Lagrangian is removed once the constraints are satisfied. Thus, the optimization is mainly guided towards minimizing the training loss (see plots in third row).

This ability to focus in reducing the loss usually come at the expense of increased density: note the slight "bounces" in model density. After reaching feasibility, models trained with dual restarts present small increases in density which are *quickly mitigated* by further growth of the multiplier. As demonstrated throughout our experiments, we can reliably achieve models that are feasible or within $(\sim 1\%)$ of the desired target level.

### G.1   Dual Restarts as Best Responses

Our proposed "dual restart" scheme is theoretically motivated as a choice of best-response from the dual player when the constraints are satisfied. Without loss of generality, we present the argument below in the case of a single inequality constraint. When there are multiple inequality constraints, the best response problem for the dual player decouples into *individual* problems for each of the Lagrange multipliers.

Given choices $[\theta, \phi]$ by the primal player, consider the optimization problem faced by the dual player:

$$\lambda_{\text{co}}^{\text{BR}}(\tilde{\theta}, \phi) = \underset{\lambda_{\text{co}} \geq 0}{\operatorname{argmax}} \, \mathfrak{L}(\tilde{\theta}, \phi, \lambda_{\text{co}}) = \underset{\lambda_{\text{co}} \geq 0}{\operatorname{argmax}} \quad \mathfrak{f}_{\text{obj}}(\tilde{\boldsymbol{\theta}}, \boldsymbol{\phi}) + \lambda_{\text{co}} \left( \mathfrak{g}_{\text{const}}(\phi) - \epsilon \right) \tag{12}$$

This is a linear optimization problem with a trivial solution: if the constraint is being satisfied $(\mathfrak{g}_{\text{const}}(\phi) - \epsilon < 0)$, then $\lambda_{\text{co}}^{\text{BR}} = 0$. If the constraint is satisfied with equality, $\lambda_{\text{co}}^{\text{BR}} = \mathbb{R}^+$. Finally, if the constraint is violated $(\mathfrak{g}_{\text{const}}(\phi) - \epsilon > 0)$, then $\lambda_{\text{co}}^{\text{BR}} = \infty$.

In summary, applying dual restarts corresponds to updating the value of the Lagrange multiplier following a best response for the dual player, regardless of the current value of the Lagrange multiplier! However, note that the same reasoning cannot be applied to the case of violated constraints: stability and overflow issues render a choice of $\infty$ for a Lagrange multiplier to be impractical for a numerical implementation.

Finally, we emphasize that while gradient ascent is an effective and simple tool for updating the Lagrange multipliers, further empirical and theoretical investigation on the influence of the dual update scheme could be beneficial.

## H   Learning Sparsity-Controlled (Wide)ResNets

ResNets have been a challenging setting for $L_0$-penalty based methods. Gale et al. [14] trained WideResNets (WRNs) [45] and ResNet50 [19] using the penalized $L_0$-regularization framework of Louizos et al. [31], and reported being unable to produce sparse residual models without significantly compromising performance.[§§] Our initial experiments on WRNs confirmed a similar behavior.

We detect two main modifications that enable us to learn WRNs and ResNets with controllable sparsity, while retaining good performance: ① increasing the learning rate of the stochastic gates, shown in Fig. 7; and ② removing the gradient contribution of the weight decay penalty towards the gates, displayed in Fig. 8.

For conciseness, we present these observations in the case of WideResNets. We adopt the two adjustments presented in this section for our experiments involving WideResNet-28-10, ResNet18 and ResNet50 models. These two simple modifications allowed us to achieve reliable controllability for (Wide)ResNets without performance degradation, just as with the MLP and LeNet architectures.

Further exploration of these modifications, and their influence in the resulting sparsity of the model are provided in Appendix I.

---

[§§]Gale et al. [14] state: *"Across hundreds of experiments, our [ResNet50] models were either able to achieve full test set performance with no sparsification, or sparsification with test set performance akin to random guessing."*; and *"Applying our weight-level $L_0$ regularization implementation to WRN produces a model with comparable training time sparsity, but with no sparsity in the test-time parameters. For models that achieve test-time sparsity, we observe significant accuracy degradation on CIFAR-10."*

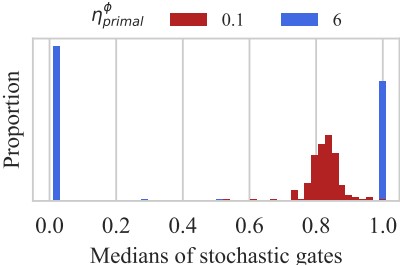

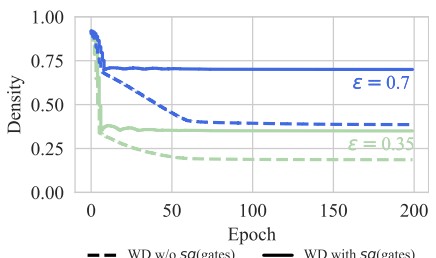

Figure 7: Distribution of gate medians for the first layer of a WRN, at the end of (penalized) training using $\lambda_{\text{pen}} = 10^{-3}$.

Figure 8: Not detaching the gradient contribution of the weight decay (WD) penalty leads to excesive sparsification.

## H.1 Sparsity in (Wide)ResNets by Tuning the Learning Rate of the Gates.

Replicating the CIFAR10 experiments of Louizos et al. [31] on WRNs using their choice of regularization parameter and learning rate for the stochastic gates, results in a distribution of the stochastic gates which does not induce sparsity in the model. Fig. 7 shows (in red) the distribution of the gate medians in the first layer of a WRN trained using a learning rate of $\eta^{\phi}_{\text{primal}} = 0.1$ for the gates parameters, as in Louizos et al. [31]. We observed that this distribution of medians did not change significantly during training. Thus, since the model has a high initial density, the gate parameters at the end of training do not induce any sparsity.

To enable the gate parameters to effectively change during training, we decoupled the learning rate of the gates $\eta^{\phi}_{\text{primal}}$, from that of the model weights $\eta^{\tilde{\theta}}_{\text{primal}}$. Fig. 7 illustrates how increasing $\eta^{\phi}_{\text{primal}}$ from 0.1 to 6 leads to a drastically different distribution for the gate medians. This simple change results in a distribution of medians which exhibits the desired concentration behavior: a non-negligible proportion of gates have a median of zero, and are therefore inactive (see Appendix A.1).

We adopt this learning rate adjustment in all our WRN experiments. Table 12 presents the performance of WRNs trained using different values of $\eta^{\phi}_{\text{primal}}$, for both the constrained and penalized settings. Note how the models using a higher learning rate for the gates parameters successfully achieve sparsity *without any compromise in performance*.

## H.2 A Loophole in Weight Decay Leads to Excessive Regularization.

This section includes the adjustment to the learning rate of the gates presented above. We noticed a systematic over-sparsification behavior in WRNs when solving the constrained formulation: the constraint is satisfied, well beyond the prescribed density level.

This issue is illustrated in Fig. 8. Dashed lines correspond to the weight decay from Louizos et al. [31] and solid lines correspond to our method with the modified weight decay as in Eq. (13). Experimental details for CIFAR-10 experiments are provided in Appendix J.4.

We identified the cause of this phenomenon to be the $L_2$ weight decay term in the training objective of WRNs from Louizos et al. [31] (see Section 2). This penalty term depends both on the probability of gates being active $\pi_j = \mathbb{P}[z_j \neq 0]$, as well as the norm of the signed magnitudes $\tilde{\theta}_j^2$. Reducing the value of this penalty could be achieved by turning off gates, such that the contribution of their associated magnitudes is ignored. This behavior is undesirable for the purpose of controllability.

We propose to restrict the effect of the weight decay penalty to $\tilde{\theta}$, as a way to reduce the parameter magnitudes, and keep the gates deactivation under the *sole influence* of the constraint violation term. We achieve this by stopping (also known as detaching) the gradients from propagating through the gate-dependent terms in the $L_2$ norm:

$$\mathbb{E}_{\mathbf{z}|\phi}\left[\|\hat{\boldsymbol{\theta}}\|_2^2\right] = \sum_{j=1}^{|\boldsymbol{\theta}|} \texttt{stop-grad}(\pi_j)\, \tilde{\theta}_j^2. \tag{13}$$

Fig. 8 shows the effect of this simple adjustment when applying a model-wise constraint on a WRN for the CIFAR10 dataset. Note how, removing the influence of weight decay on the gate parameters allows us to reliably achieve the desired target density, without over-sparsifying the model.

## I Test-Time Gates and Influence of Learning Rate and Weight Decay

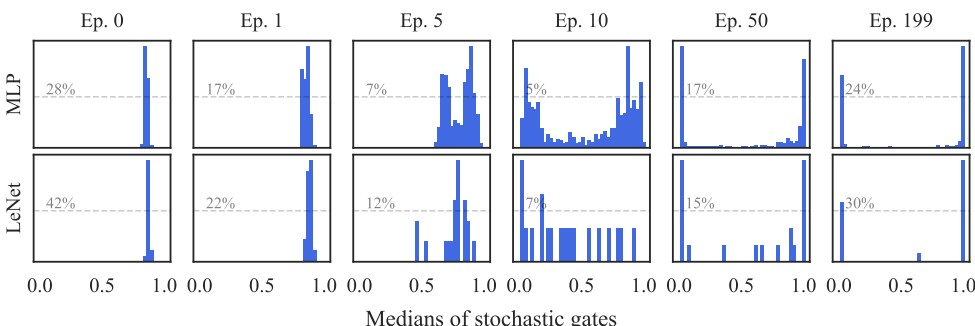

Figure 9: Histograms of *medians* of the gates in first layer of MLP and LeNet models. These models were trained on MNIST with a layer-wise target density of 70%. Note the transition from highly concentrated around a fractional value of 0.8 at the first epochs, towards an approximately binary distribution peaking at 0 and 1 at the last epoch.

Recall that we make the gates "freeze" the gates at their medians to obtain a deterministic model to evaluate on unseen data (Appendix A.1). We analyze the behavior of gates at test-time by considering histograms of their medians across specific layers of models.[¶] This section provides further empirical evidence to support the hypothesis presented in Appendix H (see Appendix J.4 for experiment setup).

Fig. 9 contains histograms for the first layer of an MLP and a LeNet trained to classify MNIST digits. These correspond to a fully connected and a convolutional layer, respectively. Sparsity requirements are specified via layer-wise constraints with a target density of 70% on both cases. Experimental details for these runs match those presented in Appendix J. Test-time gate medians are measured at the *end* of the training epochs shown in the panel titles.

At initialization, the distributions of different gates are highly similar and yield closely packed medians. These are mostly fractional in value, away from being 0 or 1. As training progresses, the medians drift apart. Histograms peak at 0 and 1 as of the 50th epoch. As expected, approximately $30\% = (100 - 70)\%$ of gates are inactive at the end of training.

Fig. 10 contains histograms of the test-time gates associated with the first convolutional layer of various WRNs-28-10 trained on CIFAR10. The proportion of gates at zero specifies the amount of inactive gates, while bars in $(0, 1]$ correspond to active gates. Since all histograms have different scales, we provide a reference line for each, corresponding to *half* the height of the tallest bar in said histogram. For example, if the dashed line is labeled as 15%, then the highest bar in that histogram corresponds to 30%.

We consider 6 configurations spanning: three learning rates for the gate parameters $\phi$; and whether or not to remove the gradient contribution of the weight decay towards the gates (see Appendix H). All experiments use the same initialized model. Sparsity requirements are specified via layer-wise constraints with a target density of 70% on all cases. Measurements of the gate medians are made at the *end* of each of the presented epochs.

Medians do not drift apart noticeably when employing $\eta^{\phi}_{\text{primal}} = 0.1$. In addition, they maintain fractional values (i.e., remain away from 0 and 1). Given our protocol for choosing gates at test-time in Appendix A.1, this would lead to a "fully dense" test-time network, which violates the required sparsity constraints by a wide margin. Similar to Gale et al. [14], we observed that when a WRN trained with $\eta^{\phi}_{\text{primal}} = 0.1$ achieved *any* significant degree of sparsity, it led to a performance akin

---

[¶]Note the subtle detail that these histograms are based on a statistic (the median) of a probability distribution and do not represent distributions of gates by themselves.

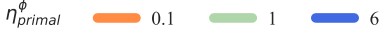

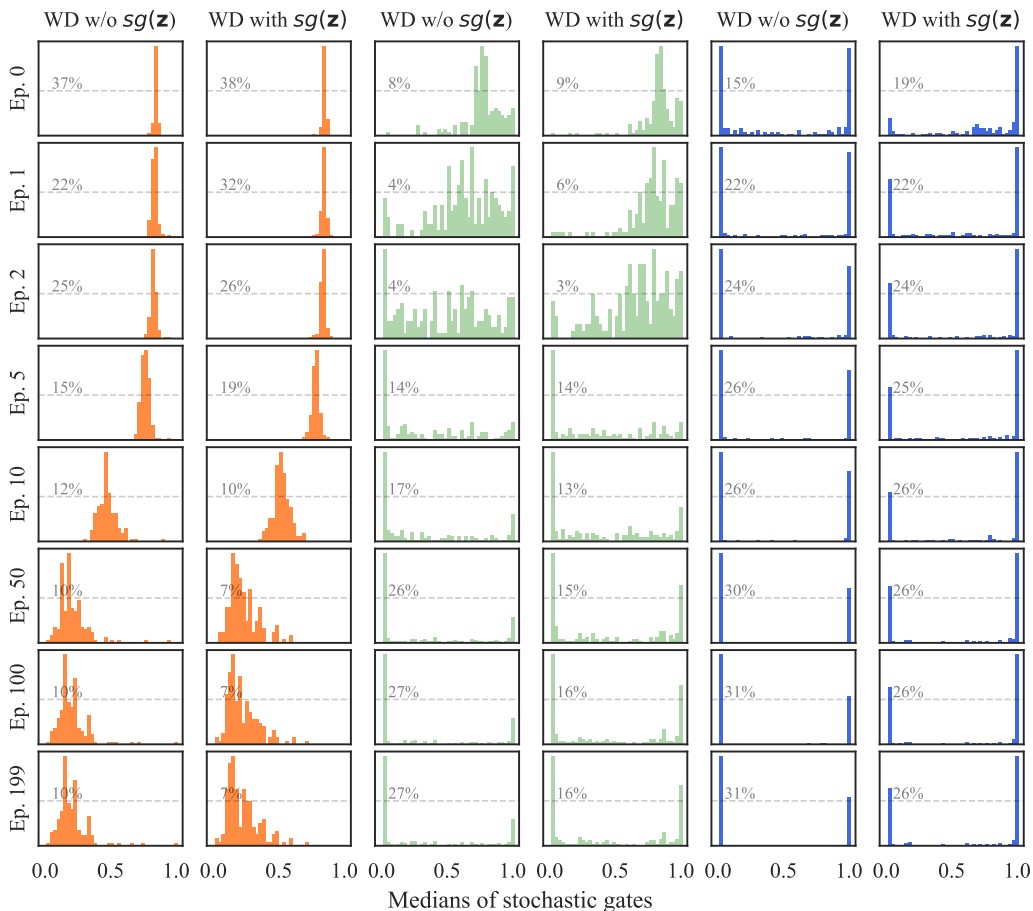

Figure 10: Histograms of gate medians for the first convolutional layer of WRN models trained on CIFAR10 under 70% layer-wise density constraints. Configurations comprise three learning rates $\eta^{\phi}_{\mathrm{primal}}$, and removing or not the gradient contribution of the weight decay towards the gate updates.

to random guessing. Note that, unsurprisingly, the bulk of the medians gets closer to zero: this is a consequence of the model aiming to satisfy the constraint on the expectation of the $L_0$-norm.

The behavior when training with $\eta^{\phi}_{\mathrm{primal}} = 1$ and $\eta^{\phi}_{\mathrm{primal}} = 6$ stands in clear contrast to that of $\eta^{\phi}_{\mathrm{primal}} = 0.1$. The former two resemble more closely the dynamics observed in Fig. 9, where medians disperse and tend to accumulate and saturate at 0 or 1. Unsurprisingly, this accumulation happens more quickly for experiments with $\eta^{\phi}_{\mathrm{primal}} = 6$. This setting also has the smallest proportion of fractional medians at the end of training. Note that in the case of $\eta^{\phi}_{\mathrm{primal}} = 1$, longer training could result in a similar outlook to that of $\eta^{\phi}_{\mathrm{primal}} = 6$. Choosing $\eta^{\phi}_{\mathrm{primal}} > 1$ in our experiments led to better performance in terms of test-time error.

Finally, note how[***] removing the gradient contribution of the weight decay towards the gates (WD **with** $\mathrm{sg}(z)$) yields yields test-time models which have approximately 30% of their parameters *inactive*, as desired. Not removing this contribution (WD **without** $\mathrm{sg}(z)$) results in over-sparsification: the distribution of gate medians is shifted towards zero for all learning rate choices. For example, consider the panels at the last epoch for $\eta^{\phi}_{\mathrm{primal}} = 1$ and $\eta^{\phi}_{\mathrm{primal}} = 6$. The experiments *with* detaching result in models whose sparsity is close to the desired sparsity level (peak at 0 close to 30%). In contrast, when not detaching, the rate of *inactive* gates is around 60%, twice as sparse as required.

---

[***]Except for the undesirable setting $\eta^{\phi}_{\mathrm{primal}} = 0.1$, as discussed in Appendix H.

# J  Experimental Details

Our implementation is developed in Python 3.8, using Pytorch 1.11 [39] and the Cooper constrained optimization library [15]. We provide scripts to replicate the experiments in this paper at: https://github.com/gallego-posada/constrained_sparsity.

All our models use ReLU activations. Throughout our experiments ① we decouple the learning rates used for the weights and gates of the network, and ② when employing weight decay, we remove the gradient contribution of the penalty towards the gates, as explained in Appendix H.

## J.1  Model Statistics

Table 4 describes the different architectures used throughout our experiments in terms of their total number of trainable parameters and the computational cost involved in a *forward* calculation in MACs (multiply-accumulate operations). We also provide details on the input size and the number of training examples for their respective datasets.

Table 4: Count of parameters and MACs for all architectures used in this paper, along with dimension and number of training examples.

| Model Type | Parameters | MACs | Input size | Train set size | Dataset |
|---|---|---|---|---|---|
| MLP | 266k | 267k | (28, 28) | 50k | MNIST |
| LeNet | 431k | 2,327M | (28, 28) | 50k | MNIST |
| WideResNet-28-10 | 36.5M | 5,959M | (3, 32, 32) | 50k | CIFAR-10/100 |
| ResNet18 | 11.3M | 6,825M | (3, 64, 64) | 100k | TinyImageNet |
| ResNet50 | 25.5M | 4,120M | (3, 224, 224) | 1.2M | ImageNet |

## J.2  Dual Optimizer

Note that the constraint functions considered throughout this work involve expectations but can be computed in closed-form based on the parameters of the gates (Appendix A). Therefore, the computation of constraint violations is deterministic. We employ gradient *ascent* on the Lagrange multipliers. We initialize all Lagrange multipliers at zero. Details on the chosen dual learning rate, along with the use of dual restarts are provided for each experiment below.

## J.3  MNIST

Following Louizos et al. [31], our experiments on MNIST classification consider two different architectures: i) an MLP with 2 hidden layers with 300 and 100 units respectively, and ii) a LeNet-5 network, consisting on convolutional layers of 20 and 50 output channels, each succeeded by a max-pooling layer with stride 2; followed by two fully connected layers of 800 and 500 input dimensions. All fully connected layers in these models use input neuron sparsity, and all convolutional layers (for LeNet models) employ output feature map sparsity.

Table 5: Default configurations for MLP and LeNet5 experiments on MNIST.

| Approach | Weights | | Gates | | Lagrange Multipliers | | |
|---|---|---|---|---|---|---|---|
| | Optim. | $\eta_{\text{primal}}^{\boldsymbol{\theta}}$ | Optim. | $\eta_{\text{primal}}^{\boldsymbol{\phi}}$ | Optim. | $\eta_{\text{dual}}$ | Restarts |
| Constrained | Adam | $7 \cdot 10^{-4}$ | Adam | $7 \cdot 10^{-4}$ | Grad. Ascent | $10^{-3}$ | Yes |
| Penalized | | | | | - | - | - |

Table 5 presents the hyper-parameters used for learning sparse MLPs and LeNets. Both cases employ the same configuration for the primal optimizer: Adam [23] with $(\beta_1, \beta_2) = (0.9, 0.99)$, as provided by default in Pytorch, with a batch size of 128. These experiments do not use weight decay.

## J.4  CIFAR-10 and CIFAR-100

We employ WideResNet-28-10 (WRN) models [45] for the tasks of classifying CIFAR-10 and CIFAR-100 images. Akin to Louizos et al. [31], the first convolutional layer in each residual block uses

output feature map sparsity, whereas the following convolutional layer and the residual connection are kept to be fully dense. This model counts with 12 sparsifiable convolutional layers.

Table 6 presents the hyper-parameters used for learning sparse WRNs. We use SGD with a momentum coefficient of 0.9 for the weights and gates. We use a batch size of 128 for 200 epochs. The primal learning rate is multiplied by 0.2 at 60, 120 and 160 epochs. This mimics the training procedure of Zagoruyko and Komodakis [45] and Louizos et al. [31]. These experiments use $\rho_{\text{init}} = 0.3$ (see Appendix A.2).

Table 6: Default configurations for WideResNet-28-10 experiments on CIFAR-10 and CIFAR-100.

| Approach | Weights | | Gates | | Lagrange Multipliers | | | Weight decay |
|---|---|---|---|---|---|---|---|---|
| | Optim. | $\eta^{\theta}_{\text{primal}}$ | Optim. | $\eta^{\phi}_{\text{primal}}$ | Optim. | $\eta_{\text{dual}}$ | Restarts | coefficient |
| Constrained | SGDM | 0.1 | SGDM | 6 | Grad. Ascent | $7 \cdot 10^{-4}$ | Yes | $5 \cdot 10^{-4}$ |
| Penalized | | | | | - | - | - | |

## J.5  TinyImageNet

We employ ResNet18 models for the task of classifying TinyImageNet [28] images. The model's initial convolutional and final fully connected layers are kept fully dense. The residual connection of each `BasicBlock` in the model is kept fully dense, while all other convolutional layers employ output feature-map sparsity. This model thus counts with 16 sparsifiable convolutional layers.

Table 7 presents the hyper-parameters used for learning sparse ResNet18s. We use SGD with a momentum coefficient of 0.9 for the weights and gates. We use a batch size of 100 for 120 epochs. The learning rate of the weights $\eta^{\tilde{\theta}}_{\text{primal}}$ is multiplied by 0.1 at 30, 60 and 90 epochs. This mimics the training procedure of previous works [25]. This experiment uses $\rho_{\text{init}} = 0.3$ (see Appendix A.2).

Table 7: Default configurations for ResNet18 experiments on TinyImageNet.

| Approach | Grouping | Weights | | Gates | | Lagrange Multipliers | | | Weight decay |
|---|---|---|---|---|---|---|---|---|---|
| | | Optim. | $\eta^{\theta}_{\text{primal}}$ | Optim. | $\eta^{\phi}_{\text{primal}}$ | Optim. | $\eta_{\text{dual}}$ | Restarts | coefficient |
| Constrained | Model Layer | SGDM | 0.1 | SGDM | 1 | Grad. Ascent | $8 \cdot 10^{-4}$ $1 \cdot 10^{-4}$ | Yes | $5 \cdot 10^{-4}$ |
| Penalized | Model Layer | | | | | | - - | - - | |

## J.6  ImageNet

We employ ResNet50 models for the task of classifying ImageNet [7] images. The model's initial convolutional and final fully connected layers are kept fully dense. The residual connection of each `Bottleneck` block in the model is kept fully dense, while all other convolutional layers employ output feature-map sparsity. This model thus counts with 48 sparsifiable convolutional layers.

Due to the high computational cost of ImageNet experiments, and the tunability issues of the penalized method, we do not perform penalized experiments for this dataset.

Table 8: Default configurations for ResNet50 experiments on ImageNet.

| Approach | Grouping | Weights | | Gates | | Lagrange Multipliers | | | Weight decay |
|---|---|---|---|---|---|---|---|---|---|
| | | Optim. | $\eta^{\theta}_{\text{primal}}$ | Optim. | $\eta^{\phi}_{\text{primal}}$ | Optim. | $\eta_{\text{dual}}$ | Restarts | coefficient |
| Constrained | Model Layer | SGDM | 0.1 | SGDM | 1 | Grad. Ascent | $3 \cdot 10^{-4}$ $3 \cdot 10^{-5}$ | Yes | $10^{-4}$ |

Table 8 presents the hyper-parameters used for learning sparse ResNet50s. SGD with a momentum coefficient of 0.9 is used for the weights and the gates. We use a batch size of 256 for 90 epochs. The learning rate of the weights $\eta^{\tilde{\theta}}_{\text{primal}}$ is multiplied by 0.1 at 30 and 60 epochs. This mimics the training procedure of previous works [40].

**Initialization of the gates**

As discussed in Appendix A.2, the choice of hyperparameter $\rho_{\text{init}}$ affects the initial sparsity of the network. For their experiments using WideResNet models on the CIFAR10/100 datasets, Louizos et al. [31] chose $\rho_{\text{init}} = 0.3$. When executing our experiments with ResNet50 models on ImageNet,

we noticed that this hyper-parameter of the $L_0$-reparametrization can have a significant impact in the behavior of the model throughout training.

Fig. 11 shows the training and validation error of two different runs at a target density of 70% with $\rho_{\text{init}} = 0.3$ and $\rho_{\text{init}} = 0.05$. Although both runs achieve the desired sparsity target, the model initialized at $\rho_{\text{init}} = 0.05$ outperforms the model initialized with $\rho_{\text{init}} = 0.3$. This performance improvement persists throughout training, resulting in a final model with a better validation error.

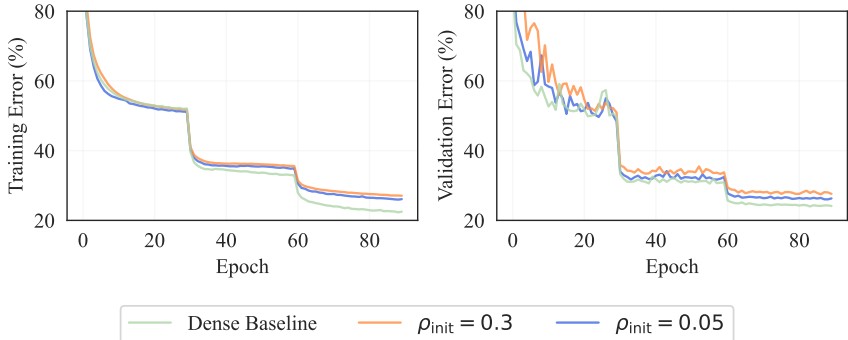

Figure 11: Effect of the initialization hyper-parameter $\rho_{\text{init}}$ for training ResNet50 models on ImageNet.

Recall that we consider fixed values for the parameters $\beta$, $\gamma$ and $\zeta$ associated with the concrete distribution, as detailed in Table 3. With these values, Eq. (11) yields that the initial $L_0$-density of the model initialized with $\rho_{\text{init}} = 0.3$ is $92.03\%$, while for $\rho_{\text{init}} = 0.05$, the $L_0$-density of the initial model is $98.95\%$. This means that the case $\rho_{\text{init}} = 0.3$ starts from a (in expectation) $\sim 7\%$ sparser model; thus unnecessarily restring the model capacity at the beginning of training. This is consistent with the behavior displayed in Fig. 11: the model initialized with $\rho_{\text{init}} = 0.05$ follows more closely the validation performance of a dense baseline (i.e. a standard ResNet50 without gates).

Considering this behavior, for all the ImageNet results reported below we use the lower value of $\rho_{\text{init}} = 0.05$. For other models and datasets, we employ the value used in Louizos et al. [31] for ease of comparison.

### J.7 Magnitude Pruning Comparison on ImageNet

We compare the performance of our in-training sparsity method with structured magnitude pruning. We start from a pre-trained ResNet50 model from Pytorch (`torchvision.models.resnet50`) and apply layer-wise pruning following the procedure of Li et al. [29] to *the same layers* that were sparsifiable for our ImageNet models, described in Appendix J.6. After performing magnitude pruning, we fine-tune the models for 20 epochs on the ImageNet dataset, using SGD with momentum of 0.9 and a constant learning rate of 0.001. This matches the fine-tuning setting of Li et al. [29].

### J.8 Sparsity Collapse

Some of the results for the penalized method presented in Appendix K are labeled as "Failed due to sparsity collapse". This means that the penalty factor was too high and resulted in all the gates of a layer being turned off.

## K   Comprehensive Experimental Results

In this section we provide complete results for all experiments, whose hyper-parameter configurations can be found in Appendix J. To make the navigation of these results easier, we repeat some of the tables and figures provided in the main paper. These repetitions are clearly marked in the caption of the corresponding resource.

## K.1 MNIST

Table 9: Achieved density levels and performance for sparse MLP and LeNet5 models trained on MNIST for 200 epochs. Metrics aggregated over 5 runs. [†]Results by Louizos et al. [31] with $N$ representing the training set size (see Appendix C). *This table is the same as Table 1. We repeat it here for the reader's convenience.*

| Architecture | Grouping | Method | Hyper-parameters | Pruned architecture | Val. Error (%) best | Val. Error (%) at 200 epochs avg $\pm$ 95% CI |
|---|---|---|---|---|---|---|
| MLP 784-300-100 | Model | Pen. | [†]$\lambda_{pen} = 0.1/N$ | 219-214-100 | 1.4 | – |
| | | Const. | $\epsilon = 33\%$ | 198-233-100 | 1.36 | $1.77 \pm 0.08$ |
| | Layer | Pen. | [†]$\lambda_{pen} = [0.1, 0.1, 0.1]/N$ | 266-88-33 | 1.8 | – |
| | | Const. | $\epsilon = [30\%, 30\%, 30\%]$ | 243-89-29 | 1.58 | $2.19 \pm 0.12$ |
| LeNet5 20-50-800-500 | Model | Pen. | [†]$\lambda_{pen} = 0.1/N$ | 20-25-45-462 | 0.9 | – |
| | | Const. | $\epsilon = 10\%$ | 20-21-34-407 | 0.56 | $1.01 \pm 0.05$ |
| | Layer | Pen. | [†]$\lambda_{pen} = [10, 0.5, 0.1, 0.1]/N$ | 9-18-65-25 | 1.0 | – |
| | | Const. | $\epsilon = [50\%, 30\%, 70\%, 10\%]$ | 10-14-224-29 | 0.7 | $0.91 \pm 0.05$ |

**MLPs**

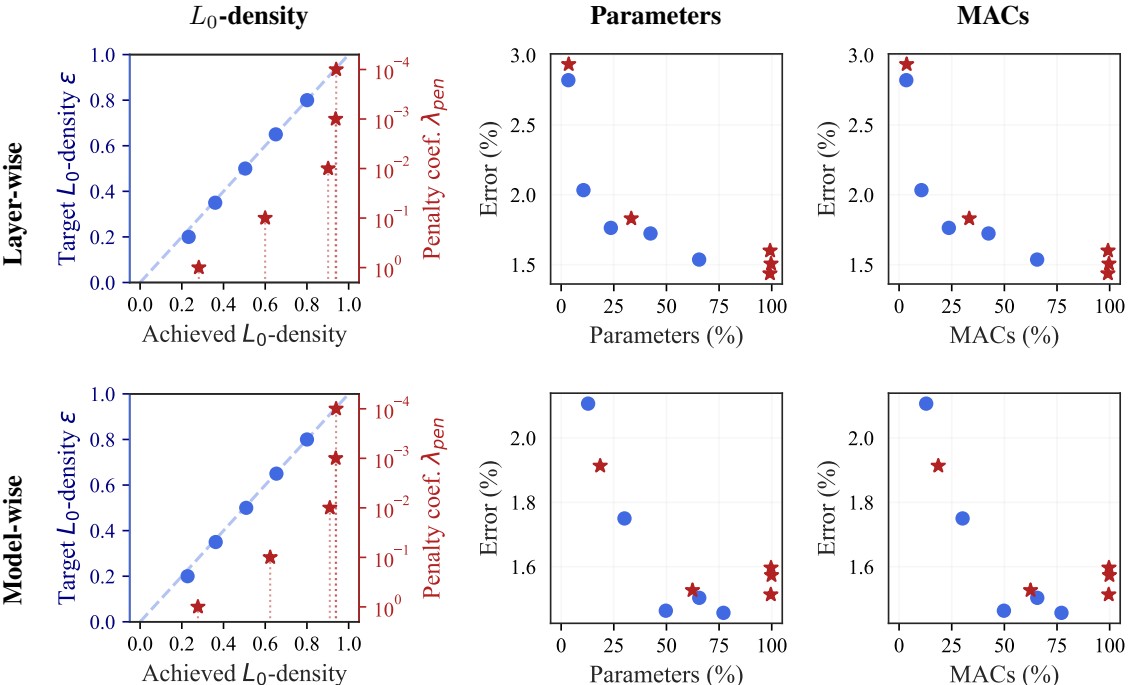

Figure 12: Training sparse MLP models on MNIST.

Table 10: Achieved density levels and performance for sparse MLP models trained on MNIST for 200 epochs. Metrics aggregated over 5 runs.

| Method | Hyper-params | $L_0$-density (%) | Params (%) | MACs (%) | Val. Error (%) best | at 200 epochs |
|---|---|---|---|---|---|---|
| Constrained $g \in [1:3]$ *Layer-wise* | $\epsilon_g = 20\%$ | $23.28 \pm 0.31$ | $3.41 \pm 0.16$ | $3.43 \pm 0.16$ | 1.65 | $2.82 \pm 0.31$ |
| | $\epsilon_g = 35\%$ | $36.02 \pm 0.08$ | $10.59 \pm 0.17$ | $10.62 \pm 0.17$ | 1.66 | $2.03 \pm 0.13$ |
| | $\epsilon_g = 50\%$ | $50.42 \pm 0.04$ | $23.60 \pm 0.14$ | $23.63 \pm 0.14$ | 1.58 | $1.76 \pm 0.15$ |
| | $\epsilon_g = 65\%$ | $65.07 \pm 0.03$ | $42.43 \pm 0.51$ | $42.46 \pm 0.51$ | 1.42 | $1.72 \pm 0.01$ |
| | $\epsilon_g = 80\%$ | $80.08 \pm 0.07$ | $65.53 \pm 0.76$ | $65.55 \pm 0.76$ | 1.38 | $1.54 \pm 0.05$ |
| Penalized $g \in [1:3]$ *Layer-wise* | $\lambda_{pen}^g = 1$ | $28.16 \pm 0.97$ | $3.58 \pm 0.51$ | $3.60 \pm 0.51$ | 2.62 | $2.93 \pm 0.09$ |
| | $\lambda_{pen}^g = 0.1$ | $59.97 \pm 0.09$ | $33.29 \pm 0.06$ | $33.32 \pm 0.06$ | 1.37 | $1.83 \pm 0.07$ |
| | $\lambda_{pen}^g = 0.01$ | $90.18 \pm 0.22$ | $99.15 \pm 0.94$ | $99.15 \pm 0.94$ | 1.29 | $1.44 \pm 0.13$ |
| | $\lambda_{pen}^g = 0.001$ | $93.77 \pm 0.10$ | $99.23 \pm 0.22$ | $99.23 \pm 0.22$ | 1.33 | $1.60 \pm 0.10$ |
| | $\lambda_{pen}^g = 0.0001$ | $94.04 \pm 0.23$ | $99.67 \pm 0.38$ | $99.67 \pm 0.38$ | 1.26 | $1.51 \pm 0.04$ |
| Constrained *Model-wise* | $\epsilon = 20\%$ | $22.77 \pm 0.17$ | $12.80 \pm 0.41$ | $12.87 \pm 0.41$ | 1.40 | $2.11 \pm 0.05$ |
| | $\epsilon = 35\%$ | $36.25 \pm 0.06$ | $30.07 \pm 0.39$ | $30.16 \pm 0.39$ | 1.37 | $1.75 \pm 0.11$ |
| | $\epsilon = 50\%$ | $50.89 \pm 0.12$ | $49.71 \pm 0.27$ | $49.78 \pm 0.27$ | 1.20 | $1.46 \pm 0.16$ |
| | $\epsilon = 65\%$ | $65.37 \pm 0.01$ | $65.57 \pm 0.22$ | $65.62 \pm 0.22$ | 1.27 | $1.50 \pm 0.03$ |
| | $\epsilon = 80\%$ | $80.02 \pm 0.05$ | $77.08 \pm 0.86$ | $77.11 \pm 0.86$ | 1.16 | $1.46 \pm 0.16$ |
| Penalized *Model-wise* | $\lambda_{pen} = 1$ | $27.73 \pm 0.15$ | $18.54 \pm 0.51$ | $18.62 \pm 0.52$ | 1.48 | $1.91 \pm 0.07$ |
| | $\lambda_{pen} = 0.1$ | $62.38 \pm 0.16$ | $62.37 \pm 0.3$ | $62.43 \pm 0.3$ | 1.30 | $1.53 \pm 0.13$ |
| | $\lambda_{pen} = 0.01$ | $90.98 \pm 0.18$ | $99.56 \pm 0.43$ | $99.56 \pm 0.43$ | 1.34 | $1.51 \pm 0.11$ |
| | $\lambda_{pen} = 0.001$ | $93.81 \pm 0.17$ | $99.89 \pm 0.22$ | $99.89 \pm 0.22$ | 1.24 | $1.57 \pm 0.09$ |
| | $\lambda_{pen} = 0.0001$ | $93.99 \pm 0.06$ | $99.67 \pm 0.38$ | $99.67 \pm 0.38$ | 1.32 | $1.60 \pm 0.17$ |

**LeNet5**

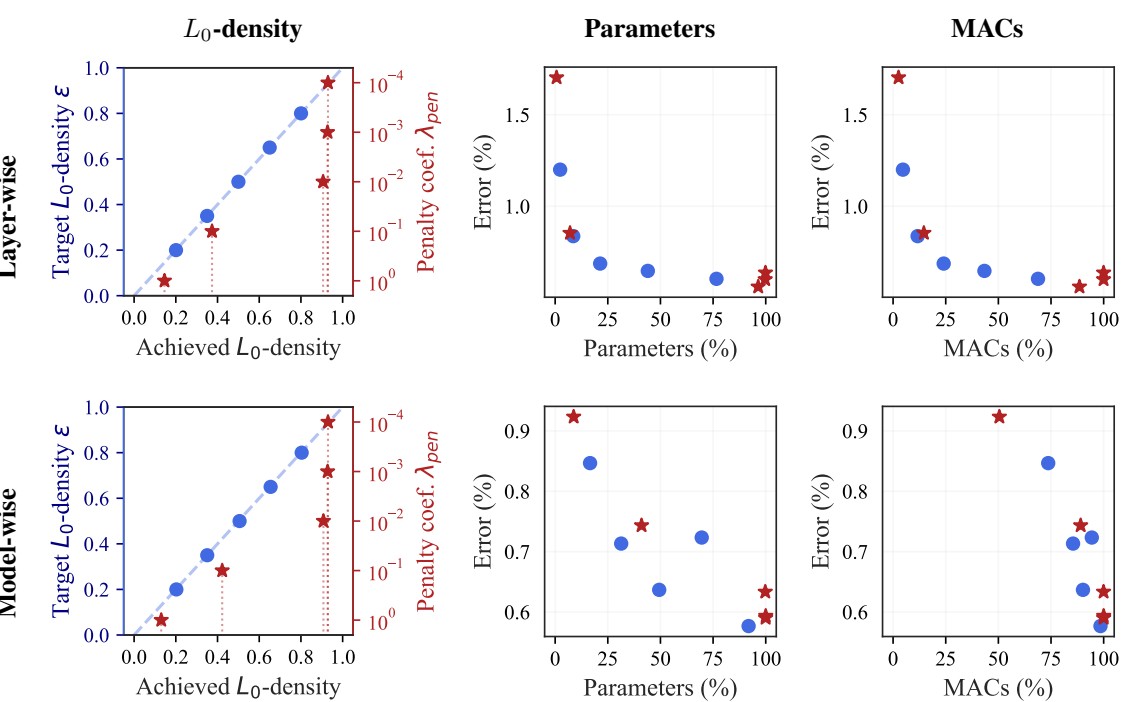

Figure 13: Training sparse LeNet models on MNIST.

Table 11: Achieved density levels and performance for sparse LeNet models trained on MNIST for 200 epochs. Metrics aggregated over 5 runs.

| Method | Hyper-params | $L_0$-density (%) | Params (%) | MACs (%) | Val. Error (%) best | at 200 epochs |
|---|---|---|---|---|---|---|
| Constrained $g \in [1:4]$ *Layer-wise* | $\epsilon_g = 20\%$ | $20.09 \pm 0.07$ | $2.36 \pm 0.14$ | $4.68 \pm 0.68$ | 0.81 | $1.20 \pm 0.08$ |
| | $\epsilon_g = 35\%$ | $35.01 \pm 0.01$ | $8.59 \pm 0.03$ | $11.65 \pm 1.15$ | 0.69 | $0.84 \pm 0.08$ |
| | $\epsilon_g = 50\%$ | $50.02 \pm 0.01$ | $21.38 \pm 0.77$ | $24.03 \pm 1.64$ | 0.56 | $0.69 \pm 0.07$ |
| | $\epsilon_g = 65\%$ | $65.02 \pm 0.01$ | $44.01 \pm 2.16$ | $43.38 \pm 2.39$ | 0.53 | $0.65 \pm 0.09$ |
| | $\epsilon_g = 80\%$ | $80.04 \pm 0.02$ | $76.62 \pm 0.99$ | $68.83 \pm 1.40$ | 0.45 | $0.60 \pm 0.04$ |
| Penalized $g \in [1:4]$ *Layer-wise* | $\lambda_{pen}^g = 1$ | $14.58 \pm 0.44$ | $0.67 \pm 0.13$ | $2.60 \pm 0.17$ | 0.46 | $1.7 \pm 0.14$ |
| | $\lambda_{pen}^g = 0.1$ | $37.38 \pm 0.49$ | $7.06 \pm 0.36$ | $14.62 \pm 1.68$ | 0.54 | $0.85 \pm 0.02$ |
| | $\lambda_{pen}^g = 0.01$ | $90.63 \pm 0.20$ | $96.43 \pm 0.83$ | $88.52 \pm 7.17$ | 0.40 | $0.56 \pm 0.03$ |
| | $\lambda_{pen}^g = 0.001$ | $92.77 \pm 0.06$ | $99.75 \pm 0.12$ | $99.95 \pm 0.02$ | 0.47 | $0.60 \pm 0.05$ |
| | $\lambda_{pen}^g = 0.0001$ | $92.96 \pm 0.04$ | $99.87 \pm 0.12$ | $99.98 \pm 0.02$ | 0.47 | $0.64 \pm 0.06$ |
| Constrained *Model-wise* | $\epsilon = 20\%$ | $20.27 \pm 0.30$ | $16.58 \pm 0.73$ | $73.67 \pm 0.14$ | 0.54 | $0.85 \pm 0.09$ |
| | $\epsilon = 35\%$ | $35.06 \pm 0.09$ | $31.36 \pm 0.27$ | $85.47 \pm 0.85$ | 0.55 | $0.71 \pm 0.03$ |
| | $\epsilon = 50\%$ | $50.58 \pm 0.09$ | $49.41 \pm 0.29$ | $90.18 \pm 0.94$ | 0.44 | $0.64 \pm 0.03$ |
| | $\epsilon = 65\%$ | $65.47 \pm 0.05$ | $69.60 \pm 1.06$ | $94.37 \pm 0.20$ | 0.44 | $0.72 \pm 0.03$ |
| | $\epsilon = 80\%$ | $80.36 \pm 0.08$ | $91.86 \pm 0.79$ | $98.49 \pm 0.15$ | 0.46 | $0.58 \pm 0.03$ |
| Penalized *Model-wise* | $\lambda_{pen} = 1$ | $13.01 \pm 0.81$ | $8.79 \pm 0.80$ | $50.47 \pm 4.74$ | 0.71 | $0.92 \pm 0.11$ |
| | $\lambda_{pen} = 0.1$ | $42.25 \pm 0.83$ | $41.03 \pm 1.12$ | $89.08 \pm 0.21$ | 0.45 | $0.74 \pm 0.04$ |
| | $\lambda_{pen} = 0.01$ | $90.78 \pm 0.20$ | $99.78 \pm 0.20$ | $99.96 \pm 0.04$ | 0.44 | $0.63 \pm 0.05$ |
| | $\lambda_{pen} = 0.001$ | $92.80 \pm 0.01$ | $100 \pm 0.00$ | $100 \pm 0.00$ | 0.43 | $0.59 \pm 0.01$ |
| | $\lambda_{pen} = 0.0001$ | $92.97 \pm 0.03$ | $99.94 \pm 0.12$ | $99.99 \pm 0.02$ | 0.45 | $0.59 \pm 0.07$ |

## K.2 CIFAR-10

Table 12: Achieved density levels and performance for sparse WideResNets-28-10 models trained on CIFAR-10 for 200 epochs. Metrics aggregated over 5 runs. [†]Result reported by Louizos et al. [31], with $N$ denoting the training set size (see Appendix C).

| Method | Hyper-params | $\eta_{primal}^\phi$ | $L_0$-density (%) | Params (%) | MACs (%) | Val. Error (%) best | at 200 epochs (avg $\pm$ 95% CI) |
|---|---|---|---|---|---|---|---|
| Penalized | [†]$\lambda_{pen} = 0.001/N$ | 0.1 | – | – | – | 3.83 | – |
| | [†]$\lambda_{pen} = 0.002/N$ | 0.1 | – | – | – | 3.93 | – |
| | $\lambda_{pen} = 0.001$ | 0.1 | $92.30 \pm 0.01$ | $99.84 \pm 0.00$ | $100 \pm 0.00$ | 4.23 | $4.56 \pm 0.18$ |
| | $\lambda_{pen} = 0.001$ | 6 | $91.19 \pm 0.16$ | $93.98 \pm 0.24$ | $91.57 \pm 0.35$ | 3.75 | $4.04 \pm 0.15$ |
| | $\lambda_{pen} = 0.002$ | 6 | $91.36 \pm 0.11$ | $94.26 \pm 0.22$ | $92.10 \pm 0.46$ | 3.62 | $4.05 \pm 0.14$ |
| Constrained $g \in [1:12]$ | $\epsilon_g = 100\%$ | 0.1 | $92.34 \pm 0.01$ | $99.84 \pm 0.00$ | $100 \pm 0.00$ | 4.18 | $4.63 \pm 0.14$ |
| | $\epsilon_g = 100\%$ | 6 | $91.63 \pm 0.27$ | $94.52 \pm 0.26$ | $92.75 \pm 0.58$ | 3.74 | $4.12 \pm 0.08$ |
| | $\epsilon_g = 70\%$ | 6 | $70.00 \pm 0.00$ | $69.87 \pm 0.20$ | $69.55 \pm 0.17$ | 3.76 | $4.10 \pm 0.16$ |

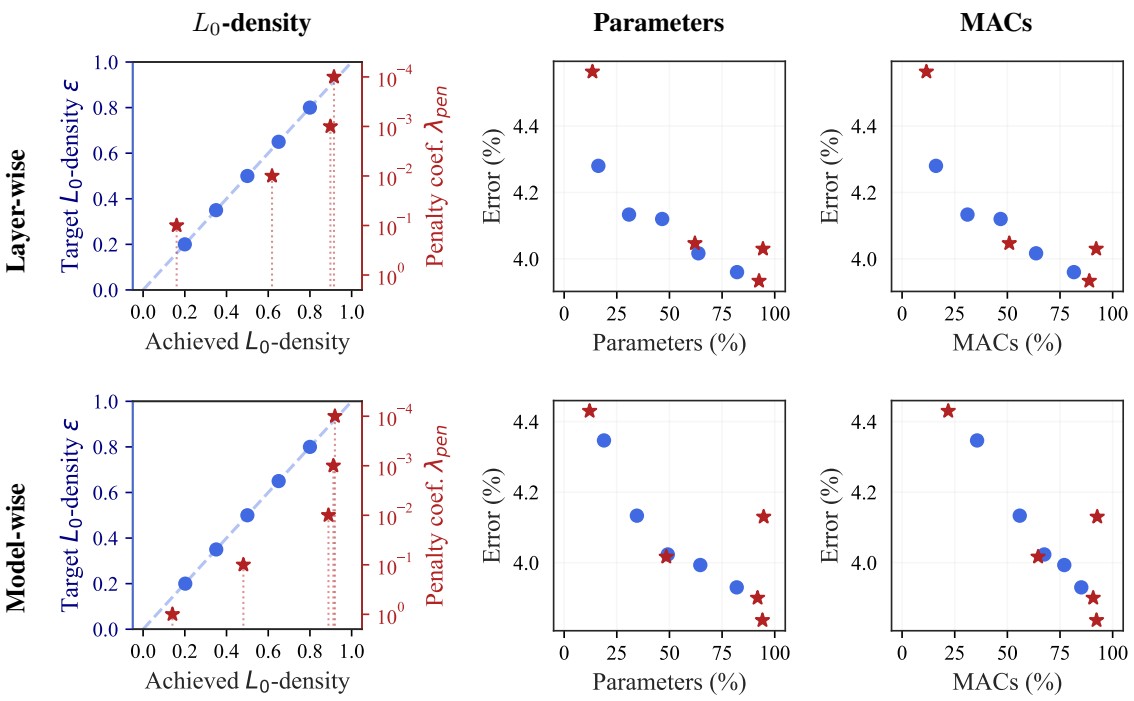

Figure 14: Training sparse WideResNet-28-10 models on CIFAR-10. *This figure is the same as Fig. 4. We repeat it here for the reader's convenience.*

Table 13: Achieved density levels and performance for sparse WideResNet-28-10 models trained on CIFAR-10 for 200 epochs. Metrics aggregated over 5 runs.

| Method | Hyper-params | $L_0$-density (%) | **Params** (%) | **MACs** (%) | **Val. Error** (%) best | at 200 epochs |
|---|---|---|---|---|---|---|
| Constrained $g \in [1:12]$ *Layer-wise* | $\epsilon_g = 20\%$ | $20.00 \pm 0.02$ | $16.24 \pm 0.20$ | $16.10 \pm 0.23$ | 4.05 | $4.28 \pm 0.16$ |
| | $\epsilon_g = 35\%$ | $34.99 \pm 0.01$ | $30.80 \pm 0.34$ | $31.05 \pm 0.23$ | 3.8 | $4.13 \pm 0.13$ |
| | $\epsilon_g = 50\%$ | $50.00 \pm 0.01$ | $46.54 \pm 0.21$ | $46.79 \pm 0.28$ | 3.87 | $4.12 \pm 0.21$ |
| | $\epsilon_g = 65\%$ | $64.99 \pm 0.01$ | $63.74 \pm 0.33$ | $63.63 \pm 0.27$ | 3.7 | $4.02 \pm 0.11$ |
| | $\epsilon_g = 80\%$ | $80.00 \pm 0.00$ | $82.10 \pm 0.32$ | $81.60 \pm 0.34$ | 3.69 | $3.96 \pm 0.15$ |
| Penalized $g \in [1:12]$ *Layer-wise* | $\lambda_{pen}^g = 1$ | —— | *Failed due to sparsity collapse* | | | —— |
| | $\lambda_{pen}^g = 0.1$ | $16.09 \pm 0.14$ | $13.43 \pm 0.11$ | $11.54 \pm 0.29$ | 4.23 | $4.56 \pm 0.09$ |
| | $\lambda_{pen}^g = 0.01$ | $61.83 \pm 0.86$ | $62.10 \pm 0.97$ | $50.91 \pm 0.62$ | 3.83 | $4.05 \pm 0.02$ |
| | $\lambda_{pen}^g = 0.001$ | $89.81 \pm 0.06$ | $92.55 \pm 0.13$ | $88.91 \pm 0.15$ | 3.7 | $3.93 \pm 0.15$ |
| | $\lambda_{pen}^g = 0.0001$ | $91.54 \pm 0.33$ | $94.41 \pm 0.48$ | $92.18 \pm 0.94$ | 3.81 | $4.03 \pm 0.05$ |
| Constrained *Model-wise* | $\epsilon = 20\%$ | $20.22 \pm 0.01$ | $18.91 \pm 0.03$ | $35.62 \pm 0.84$ | 3.95 | $4.35 \pm 0.25$ |
| | $\epsilon = 35\%$ | $35.07 \pm 0.03$ | $34.55 \pm 0.08$ | $55.87 \pm 1.23$ | 3.76 | $4.13 \pm 0.27$ |
| | $\epsilon = 50\%$ | $50.00 \pm 0.01$ | $49.21 \pm 0.33$ | $67.53 \pm 0.08$ | 3.79 | $4.02 \pm 0.06$ |
| | $\epsilon = 65\%$ | $65.01 \pm 0.01$ | $64.65 \pm 0.34$ | $77.02 \pm 0.43$ | 3.80 | $3.99 \pm 0.14$ |
| | $\epsilon = 80\%$ | $80.01 \pm 0.00$ | $81.96 \pm 0.12$ | $85.11 \pm 0.58$ | 3.83 | $3.93 \pm 0.09$ |
| Penalized *Model-wise* | $\lambda_{pen} = 1$ | $14.07 \pm 0.23$ | $12.09 \pm 0.24$ | $21.92 \pm 0.59$ | 4.23 | $4.43 \pm 0.10$ |
| | $\lambda_{pen} = 0.1$ | $48.08 \pm 2.05$ | $48.50 \pm 2.27$ | $64.58 \pm 1.57$ | 3.87 | $4.02 \pm 0.22$ |
| | $\lambda_{pen} = 0.01$ | $88.89 \pm 0.31$ | $91.85 \pm 0.47$ | $90.76 \pm 0.77$ | 3.72 | $3.90 \pm 0.18$ |
| | $\lambda_{pen} = 0.001$ | $91.24 \pm 0.20$ | $94.17 \pm 0.24$ | $92.34 \pm 0.22$ | 3.59 | $3.84 \pm 0.25$ |
| | $\lambda_{pen} = 0.0001$ | $91.98 \pm 0.42$ | $94.75 \pm 0.31$ | $92.68 \pm 0.36$ | 3.91 | $4.13 \pm 0.09$ |

### K.3 CIFAR-100

Table 14: Achieved density levels and performance for sparse WideResNets-28-10 models trained on CIFAR-100 for 200 epochs. Metrics aggregated over 5 runs. [†]Result reported by Louizos et al. [31], with $N$ denoting the training set size (see Appendix C).

| Method | Hyper-params | $\eta^{\phi}_{\text{primal}}$ | $L_0$-density (%) | Params (%) | MACs (%) | Val. Error (%) | |
|---|---|---|---|---|---|---|---|
| | | | | | | best | at 200 epochs (avg $\pm$ 95% CI) |
| Penalized | $^{\dagger}\lambda_{pen} = 0.001/N$ | 0.1 | – | – | – | 18.75 | – |
| | $^{\dagger}\lambda_{pen} = 0.002/N$ | 0.1 | – | – | – | 19.04 | – |
| | $\lambda_{pen} = 0.001$ | 0.1 | $93.20 \pm 0.01$ | $100.00 \pm 0.00$ | $100.00 \pm 0.00$ | 21.01 | $21.70 \pm 0.19$ |
| | $\lambda_{pen} = 0.001$ | 6 | $90.64 \pm 0.32$ | $90.88 \pm 0.41$ | $89.94 \pm 0.71$ | 18.51 | $19.14 \pm 0.21$ |
| | $\lambda_{pen} = 0.002$ | 6 | $90.13 \pm 0.45$ | $90.19 \pm 0.38$ | $89.52 \pm 0.57$ | 18.99 | $19.24 \pm 0.14$ |
| Constrained $g \in [1:12]$ | $\epsilon_g = 100\%$ | 0.1 | $93.20 \pm 0.01$ | $100.00 \pm 0.00$ | $100.00 \pm 0.00$ | 21.02 | $21.66 \pm 0.21$ |
| | $\epsilon_g = 100\%$ | 6 | $90.77 \pm 0.31$ | $90.99 \pm 0.25$ | $89.74 \pm 0.29$ | 18.68 | $19.08 \pm 0.16$ |
| | $\epsilon_g = 70\%$ | 6 | $69.99 \pm 0.01$ | $68.62 \pm 0.08$ | $68.59 \pm 0.22$ | 18.88 | $19.37 \pm 0.15$ |

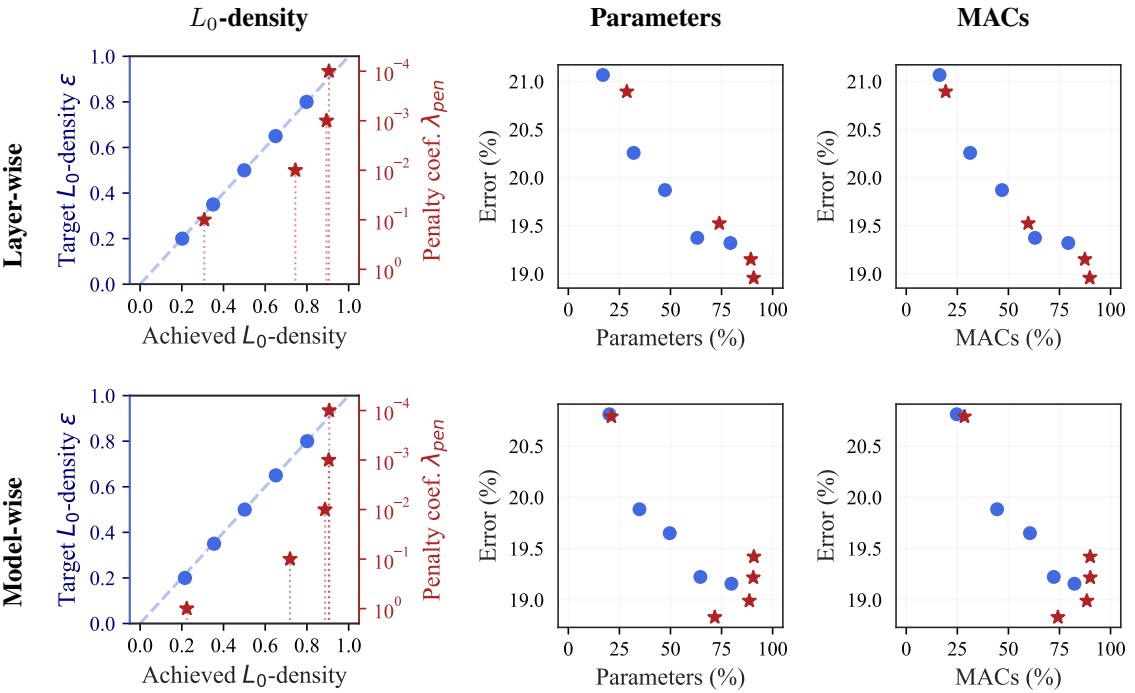

Figure 15: Training sparse WideResNet-28-10 models on CIFAR-100.

Table 15: Achieved density levels and performance for sparse WideResNet-28-10 models trained on CIFAR-100 for 200 epochs. Metrics aggregated over 5 runs.

| Method | Hyper-params | $L_0$-density (%) | Params (%) | MACs (%) | Val. Error (%) | |
|---|---|---|---|---|---|---|
| | | | | | best | at 200 epochs |
| Constrained $g \in [1:12]$ *Layer-wise* | $\epsilon_g = 20\%$ | $20.20 \pm 0.00$ | $16.94 \pm 0.06$ | $16.30 \pm 0.08$ | 20.67 | $21.07 \pm 0.19$ |
| | $\epsilon_g = 35\%$ | $35.03 \pm 0.02$ | $31.97 \pm 0.14$ | $31.27 \pm 0.08$ | 19.71 | $20.26 \pm 0.37$ |
| | $\epsilon_g = 50\%$ | $49.97 \pm 0.03$ | $47.30 \pm 0.31$ | $46.87 \pm 0.31$ | 19.58 | $19.87 \pm 0.21$ |
| | $\epsilon_g = 65\%$ | $64.99 \pm 0.02$ | $63.11 \pm 0.05$ | $63.01 \pm 0.17$ | 18.89 | $19.37 \pm 0.27$ |
| | $\epsilon_g = 80\%$ | $79.82 \pm 0.15$ | $79.33 \pm 0.30$ | $79.29 \pm 0.43$ | 18.93 | $19.32 \pm 0.21$ |
| Penalized $g \in [1:12]$ *Layer-wise* | $\lambda_{pen}^g = 1$ | —— | *Failed due to sparsity collapse* | | | —— |
| | $\lambda_{pen}^g = 0.1$ | $30.80 \pm 0.54$ | $28.63 \pm 0.52$ | $19.30 \pm 0.20$ | 20.40 | $20.89 \pm 0.29$ |
| | $\lambda_{pen}^g = 0.01$ | $74.46 \pm 0.41$ | $73.90 \pm 0.39$ | $59.69 \pm 0.36$ | 19.03 | $19.52 \pm 0.15$ |
| | $\lambda_{pen}^g = 0.001$ | $89.38 \pm 0.12$ | $89.31 \pm 0.17$ | $87.42 \pm 0.40$ | 18.72 | $19.15 \pm 0.33$ |
| | $\lambda_{pen}^g = 0.0001$ | $90.55 \pm 0.34$ | $90.74 \pm 0.34$ | $89.81 \pm 0.42$ | 18.67 | $18.96 \pm 0.22$ |
| Constrained *Model-wise* | $\epsilon = 20\%$ | $21.50 \pm 0.19$ | $20.10 \pm 0.29$ | $24.72 \pm 0.31$ | 20.42 | $20.81 \pm 0.07$ |
| | $\epsilon = 35\%$ | $35.46 \pm 0.01$ | $34.82 \pm 0.11$ | $44.46 \pm 0.76$ | 19.29 | $19.88 \pm 0.16$ |
| | $\epsilon = 50\%$ | $50.16 \pm 0.01$ | $49.63 \pm 0.09$ | $60.54 \pm 0.88$ | 19.25 | $19.65 \pm 0.17$ |
| | $\epsilon = 65\%$ | $65.08 \pm 0.04$ | $64.64 \pm 0.14$ | $72.20 \pm 0.60$ | 18.86 | $19.22 \pm 0.25$ |
| | $\epsilon = 80\%$ | $80.12 \pm 0.03$ | $79.82 \pm 0.04$ | $82.33 \pm 0.46$ | 18.91 | $19.16 \pm 0.25$ |
| Penalized *Model-wise* | $\lambda_{pen} = 1$ | $22.37 \pm 0.39$ | $21.07 \pm 0.48$ | $28.49 \pm 0.62$ | 20.34 | $20.79 \pm 0.32$ |
| | $\lambda_{pen} = 0.1$ | $71.90 \pm 0.73$ | $71.66 \pm 0.71$ | $74.22 \pm 0.43$ | 18.37 | $18.83 \pm 0.47$ |
| | $\lambda_{pen} = 0.01$ | $88.65 \pm 0.73$ | $88.60 \pm 0.80$ | $88.39 \pm 0.83$ | 18.41 | $18.99 \pm 0.37$ |
| | $\lambda_{pen} = 0.001$ | $90.46 \pm 0.23$ | $90.53 \pm 0.45$ | $90.06 \pm 0.62$ | 18.75 | $19.22 \pm 0.16$ |
| | $\lambda_{pen} = 0.0001$ | $90.70 \pm 0.33$ | $90.83 \pm 0.35$ | $90.03 \pm 0.32$ | 19.02 | $19.42 \pm 0.10$ |

## K.4 TinyImageNet

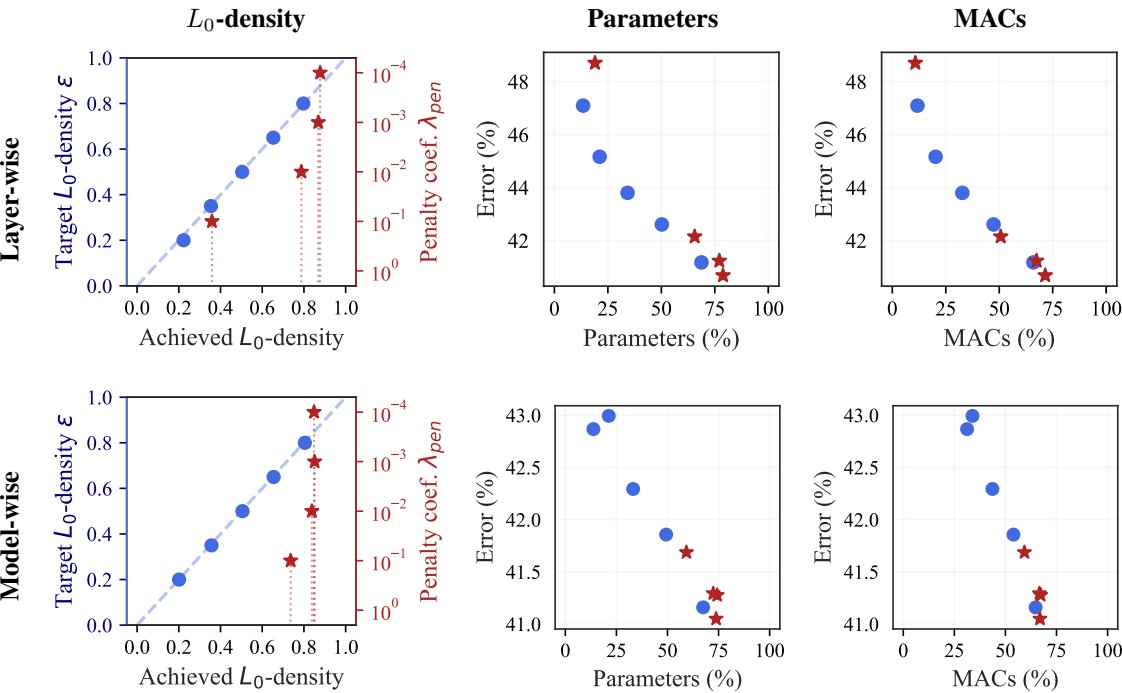

Figure 16: Training sparse ResNet18 models on TinyImageNet. *This figure is the same as Fig. 1. We repeat it here for the reader's convenience.*

Table 16: Achieved density levels and performance for sparse ResNet18 models trained on TinyImageNet for 120 epochs. Metrics aggregated over 3 runs.

| Method | Hyper-params | $L_0$-density (%) | Params (%) | MACs (%) | Val. Error (%) best | at 200 epochs |
|---|---|---|---|---|---|---|
| Constrained $g \in [1:16]$ *Layer-wise* | $\epsilon_g = 20\%$ | $22.99 \pm 0.53$ | $13.40 \pm 0.54$ | $11.75 \pm 0.45$ | 46.12 | $47.09 \pm 0.15$ |
| | $\epsilon_g = 35\%$ | $35.41 \pm 0.07$ | $21.16 \pm 0.03$ | $20.25 \pm 0.16$ | 43.87 | $45.16 \pm 0.31$ |
| | $\epsilon_g = 50\%$ | $50.40 \pm 0.04$ | $34.22 \pm 0.28$ | $32.73 \pm 0.49$ | 42.52 | $43.80 \pm 0.44$ |
| | $\epsilon_g = 65\%$ | $65.32 \pm 0.42$ | $50.22 \pm 0.54$ | $47.34 \pm 0.66$ | 41.80 | $42.61 \pm 0.10$ |
| | $\epsilon_g = 80\%$ | $79.68 \pm 0.88$ | $68.78 \pm 1.70$ | $65.98 \pm 1.53$ | 40.55 | $41.17 \pm 0.61$ |
| Penalized $g \in [1:16]$ *Layer-wise* | $\lambda_{pen}^g = 1$ | —— | *Failed due to sparsity collapse* | | | —— |
| | $\lambda_{pen}^g = 0.1$ | $35.90 \pm 0.57$ | $18.98 \pm 0.57$ | $10.77 \pm 0.90$ | 46.34 | $48.70 \pm 0.73$ |
| | $\lambda_{pen}^g = 0.01$ | $78.77 \pm 0.84$ | $65.59 \pm 1.94$ | $50.66 \pm 1.77$ | 41.36 | $42.15 \pm 0.27$ |
| | $\lambda_{pen}^g = 0.001$ | $86.87 \pm 0.38$ | $77.14 \pm 1.29$ | $67.45 \pm 3.88$ | 40.75 | $41.23 \pm 0.58$ |
| | $\lambda_{pen}^g = 0.0001$ | $87.77 \pm 0.42$ | $78.85 \pm 0.91$ | $71.49 \pm 2.36$ | 40.32 | $40.68 \pm 0.04$ |
| Constrained *Model-wise* | $\epsilon = 20\%$ | $20.11 \pm 0.03$ | $13.75 \pm 0.31$ | $31.22 \pm 1.25$ | 42.02 | $42.87 \pm 1.00$ |
| | $\epsilon = 35\%$ | $35.66 \pm 0.12$ | $21.25 \pm 0.70$ | $33.88 \pm 1.25$ | 41.28 | $42.99 \pm 0.81$ |
| | $\epsilon = 50\%$ | $50.53 \pm 0.09$ | $33.19 \pm 0.08$ | $43.65 \pm 0.94$ | 40.70 | $42.29 \pm 0.17$ |
| | $\epsilon = 65\%$ | $65.46 \pm 0.19$ | $49.34 \pm 0.55$ | $53.92 \pm 0.97$ | 41.23 | $41.86 \pm 0.54$ |
| | $\epsilon = 80\%$ | $80.45 \pm 0.19$ | $67.45 \pm 0.67$ | $64.80 \pm 1.18$ | 40.71 | $41.16 \pm 0.28$ |
| Penalized *Model-wise* | $\lambda_{pen} = 1$ | —— | *Failed due to sparsity collapse* | | | —— |
| | $\lambda_{pen} = 0.1$ | $73.64 \pm 0.72$ | $59.27 \pm 0.76$ | $59.33 \pm 2.19$ | 40.84 | $41.69 \pm 0.16$ |
| | $\lambda_{pen} = 0.01$ | $83.80 \pm 0.08$ | $72.41 \pm 0.67$ | $66.64 \pm 1.26$ | 40.33 | $41.29 \pm 0.50$ |
| | $\lambda_{pen} = 0.001$ | $85.12 \pm 0.89$ | $74.31 \pm 1.72$ | $67.30 \pm 1.62$ | 40.71 | $41.28 \pm 0.56$ |
| | $\lambda_{pen} = 0.0001$ | $84.84 \pm 0.90$ | $73.80 \pm 1.53$ | $66.91 \pm 1.65$ | 40.57 | $41.05 \pm 0.33$ |

## K.5 ImageNet

Table 17: Sparse ResNet50 models on ImageNet. "Fine-tuning" for zero epochs means *no* fine-tuning. *This table is the same as Table 2. We repeat it here for the reader's convenience.*

| Target Density | Method | $L_0$-density (%) | Params (%) | MACs (%) | Best Val. Error (%) After fine-tuning for # epochs | | | |
|---|---|---|---|---|---|---|---|---|
| | | | | | 0 | 1 | 10 | 20 |
| – | Pre-trained Baseline | 100 | [25.5M] | $[4.12 \cdot 10^9]$ | 23.90 | | | |
| $\epsilon = 90\%$ | Constrained *Model-wise* | 90.36 | 88.06 | 91.62 | **24.68** | | | |
| | Constrained *Layer-wise* | 90.58 | 87.07 | 85.97 | 24.97 | | | |
| | L1 - Mag. Prune *Layer-wise* | – | 85.94 | 84.99 | 38.74 | 25.38 | 24.69 | **24.68** |
| $\epsilon = 70\%$ | Constrained *Model-wise* | 70.78 | 64.41 | 76.50 | **25.53** | | | |
| | Constrained *Layer-wise* | 70.36 | 61.91 | 58.59 | 26.98 | | | |
| | L1 - Mag. Prune *Layer-wise* | – | 62.15 | 59.85 | 97.78 | 29.04 | 26.80 | 26.14 |
| $\epsilon = 50\%$ | Constrained *Model-wise* | 50.18 | 42.47 | 58.00 | **27.51** | | | |
| | Constrained *Layer-wise* | 50.70 | 43.15 | 38.25 | 27.89 | | | |
| | L1 - Mag. Prune *Layer-wise* | – | 43.47 | 39.76 | 99.75 | 36.21 | 29.98 | 29.16 |
| $\epsilon = 30\%$ | Constrained *Model-wise* | 30.31 | 31.81 | 42.05 | **29.65** | | | |
| | Constrained *Layer-wise* | 31.44 | 30.16 | 23.74 | 31.71 | | | |
| | L1 - Mag. Prune *Layer-wise* | – | 29.86 | 24.80 | 99.89 | 56.11 | 36.90 | 34.74 |

Here we provide further results for ResNet50 models on ImageNet, including experiments with model-wise constraints and the fine-tuned performance of the magnitude pruning method. We highlight that the controllability properties of our proposed constrained formulation extend to this large-scale setting (compare target density and $L_0$-density columns). The dense, pre-trained baseline, used as the starting point for magnitude pruning, corresponds to the `ResNet50_Weights.IMAGENET1K_V1` model made publicly available by Pytorch [39].

As expected, model-wise constraints allow for a more flexible allocation of the parameter budget throughout the network, thus leading to a better validation error. However, this flexibility can also result in models with larger memory and computational footprints. For example, with $\epsilon = 70\%$, the model learned with model-wise constraints has a very similar parameter count ($64.41\%$ vs $61.19\%$) but a significantly higher MAC count ($76.50\%$ vs $58.59\%$).

The results on magnitude pruning confirm the importance of the fine-tuning stage for this technique. The accuracy improves dramatically after a few epochs of retraining compared to the "just-pruned" model. Rather than a "pure" pruning technique, one can think of magnitude pruning as a method that, given a pretrained model, provides an *initialization* for a smaller model which needs to be trained (i.e. fine-tuned). At high density levels (e.g. 70-90%) the performance of the constrained $L_0$ formulation and fine-tuned magnitude pruning methods are located within a similar range. However, for harsher sparsity levels (30-50% density) the performance of models obtained using the constrained approach is significantly better than for magnitude pruning, even after fine-tuning.

## L  Unstructured Sparsity

In this section we demonstrate that our constrained approach transfers successfully between the structured and unstructured sparsity regimes without major modifications. We carry out experiments using MLP and convolutional models on MNIST, and ResNet18 models on TinyImageNet.

Our unstructured experiments consider one gate *per model parameter*. This means that the number of gates in the unstructured setting is much larger than in the structured one, since it scales with the total number of model parameters and not with the number of units/output maps. The $L_0$ density of a layer with unstructured sparsity corresponds to the expected number of active gates within that layer.

We compare to a magnitude pruning baseline where a dense model is pre-trained, pruned in an unstructured way and fine-tuned. Since magnitude pruning is typically applied independently at each layer, we concentrate on experiments with layer-wise constraints.

### L.1  Experimental Setting

Throughout this section, the model architectures we use for experiments with unstructured sparsity match those of structured experiments detailed in Table 4.

#### L.1.1  MNIST

Table 18 presents the hyper-parameters used for training MLP and LeNet models on MNIST with unstructured sparsity. We train using a batch size of 128 and do not apply weight decay.

Table 18: Configurations for MNIST experiments with unstructured sparsity.

| Approach | Weights | | Gates | | Lagrange Multipliers | | |
|---|---|---|---|---|---|---|---|
| | Optim. | $\eta^{\boldsymbol{\theta}}_{\text{primal}}$ | Optim. | $\eta^{\boldsymbol{\phi}}_{\text{primal}}$ | Optim. | $\eta_{\text{dual}}$ | Restarts |
| Constrained | Adam | $7 \cdot 10^{-4}$ | Adam | $1 \cdot 10^{-3}$ | Grad. Ascent | $10^{-3}$ | Yes |
| Magnitude Pruning | Adam | $7 \cdot 10^{-4}$ | - | - | - | - | - |

**Models with $L_0$ gates.** All the layers in these models have unstructured gates (one gate per weight entry and one gate per bias). The gate parameters are initialized using $\rho_{\text{init}} = 0.05$ (see Appendix A.2). We train these models for 200 epochs.

**Magnitude pruning.** For our magnitude pruning experiments we first trained a fully dense model for 200 epochs. We then apply unstructured pruning with the pre-determined target density, and retrain the resulting sparse model for another 200 epochs. We apply magnitude pruning to each of the layers in these models. Our magnitude pruning implementation keeps the biases fully dense.

### L.1.2 TinyImageNet

Table 19 presents the hyper-parameters used for learning sparse ResNet18 models on TinyImageNet. We use SGD with a momentum coefficient of 0.9 for the weights. The learning rate of the weights $\eta^{\tilde{\theta}}_{\text{primal}}$ is multiplied by 0.1 at 30, 60 and 90 epochs. We use a batch size of 100.

Table 19: Default configurations for TinyImageNet experiments with unstructured sparsity.

| Approach | Weights | | Gates | | Lagrange Multipliers | | | Target density | Weight decay |
|---|---|---|---|---|---|---|---|---|---|
| | Optim. | $\eta^{\tilde{\theta}}_{\text{primal}}$ | Optim. | $\eta^{\phi}_{\text{primal}}$ | Optim. | Restarts | $\eta_{\text{dual}}$ | | |
| Constrained | SGDM | 0.1 | Adam | $3 \cdot 10^{-2}$ | Gradient Ascent | Yes | $9 \cdot 10^{-5}$ $2 \cdot 10^{-4}$ $7 \cdot 10^{-4}$ $2 \cdot 10^{-3}$ | 20% 10% 5% 1% | $5 \cdot 10^{-4}$ |
| Magnitude Pruning | SGDM | $1 \cdot 10^{-4}$ | - | - | - | - | - | | $5 \cdot 10^{-4}$ |

**Models with $L_0$ gates.** The model's initial convolutional and final fully connected layers are kept fully dense. The residual connection of each `BasicBlock` in the model is kept fully dense, while all other convolutional layers use unstructured sparsity. This results in 16 sparsifiable convolutional layers. The gate parameters are initialized using $\rho_{\text{init}} = 0.05$ (see Appendix A.2) and optimized with Adam. We train these models for 120 epochs.

**Magnitude pruning.** For our magnitude pruning experiments we first trained a fully dense ResNet18 model for 120 epochs, using the weights learning rate schedule mentioned above. We then apply unstructured pruning with the pre-determined target density, and fine-tune the resulting sparse model for another 120 epochs using a fixed learning rate of $1 \cdot 10^{-4}$. We apply magnitude pruning to *the same layers* that were sparsifiable in models with $L_0$ gates. Our magnitude pruning implementation keeps the biases fully dense.

### L.1.3 Gates Optimizer

We originally tried the same optimization setup as with structured experiments (see Appendix J.5) for our unstructured experiments on TinyImageNet. Note that in the unstructured setting, there is a significantly larger number of gates whose parameters need to be optimized. Moreover, the influence of each individual gate on the sparsity of a layer is much smaller compared to the structured experiments. Thus the learning rates for the gates and the dual variables required to be tuned for these new unstructured tasks.

It was difficult to find a value of the gates learning rate that allowed the gates to move appropriately: (1) models trained with small learning rates would not achieve any sparsity (see Appendix A for similar behavior in the structured sparsity regime); while (2) for larger gates learning rates we observed no decrease in model density for a long portion of training, followed by a sudden drop to the target density level. However, this sudden sparsification of the network caused a significant accuracy degradation. This behavior is consistent with the observations documented by Gale et al. [14].

We hypothesize that training models with *unstructured $L_0$* gates is highly susceptible to noise. In the unstructured case the information available for determining whether a gate should be active is mediated by its single associated parameter. Since we use mini-batch estimates of the model gradients, the training signal coming from this single parameter can be very noisy. In contrast, the structured setting has lower variance since each gate aggregates information across a large group of parameters. Therefore, using an optimizer that is robust to this training noise is desirable.

We performed experiments with Adam as the optimizer for the model gates and this choice successfully delivered sparse models without breaking their predictive capacity.

Note that the experiments reported by Gale et al. [14] did not use an adaptive optimizer for the gates. Exploring whether an adaptive optimizer like Adam would be sufficient to resolve the shortcomings of the $L_0$ reparametrization framework of Louizos et al. [31] documented by Gale et al. [14] is an interesting direction for future research.

## L.2   Training Dynamics

In this section we explore the training dynamics of our proposed constrained formulation in the unstructured sparsity setting. We train a ResNet18 model on TinyImageNet with layer-wise constraints of 5% density. For other experimental settings see Appendix L.1.2.

Fig. 17 shows the overall model density, the validation error, the density for a specific layer, and the Lagrange multiplier associated with the constraint for this layer. The behavior for the chosen layer is representative of that of other layers in the model. The density at the model and layer levels decreases stably to the desired target. Note that a high sparsity of 95% is achievable without causing irreparable damage to the model accuracy.

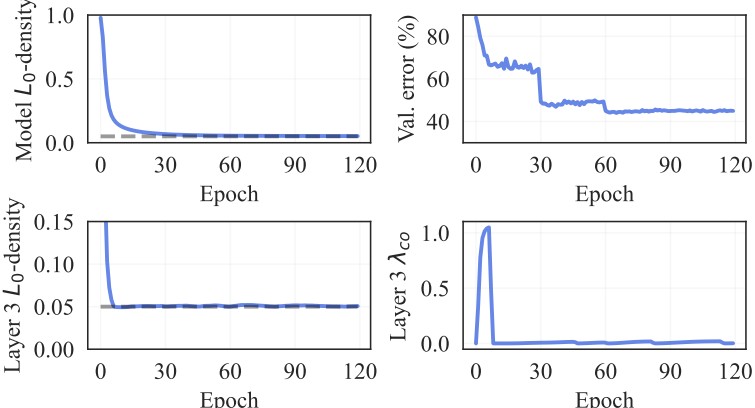

Figure 17: Training dynamics for a ResNet18 model on TinyImageNet. This unstructured sparsity experiment uses layer-wise constraints with a target density of $5\%$. *Layer 3* corresponds to the first convolutional layer of the second `BasicBlock` of the first "layer/stage" of the ResNet18 model.

Overall, the training dynamics of our unstructured experiments are qualitatively equivalent to those of the structured experiments presented in Fig. 6. This demonstrates that our constrained approach transfers successfully between structured and unstructured experiments.

## L.3   Performance Comparison

Tables 20 and 21 present the sparsity and performance statistics of MLP and LeNet models trained on MNIST at different target densities. We also include a (100%) dense model and layer-wise magnitude pruning experiments as baselines. For details on the experimental settings see Appendix L.1.

**MNIST.** Our proposed constrained approach consistently outperforms magnitude pruning even after fine tuning the magnitude pruning models for 200 epochs. Our approach reliably produces models with the desired density for experiments at $20\%$, $10\%$, and $5\%$ density. The 1% density setting is challenging for both methods.

The constrained approach achieves high accuracy, although it incurs in a small violation of the sparsity constraint. Note that we employ the same dual learning rate across all densities. More extensive tuning of the dual learning rate can resolve this unfeasibility.

On the other hand, magnitude pruning experiments achieve the desired density at 1% (by design) but drastically fail in terms of performance. Note that the accuracy for magnitude pruning does not improve to an acceptable level even after fine-tuning for a large number of epochs. This behavior can be explained by the fact that our MNIST models have some layers with very few parameters. For example, at a 1% sparsity, the first convolutional layer of the model has only 5 active parameters.

The very low number of parameters make the high-sparsity pruned models difficult to fine-tune. This observation is consistent with the poor gradient dynamics reported by Evci et al. [9] when training highly sparse networks. We would like to highlight that our proposed method applies sparsity to the same layers as magnitude pruning, yet achieves high levels of unstructured sparsity per layer with minimal accuracy reduction. This shows that very sparse models with high accuracy do exist, however they seem to be out of reach when simply fine-tuning a magnitude-pruned model.

Table 20: MLP models trained with unstructured sparsity on MNIST. Unstructured magnitude pruning is applied independently at each layer to retain the desired target density. "Fine-tuning" for zero epochs means *no* fine-tuning. Metrics averaged across 3 runs.

| Target Density $g \in [1:3]$ | Method | $L_0$-**density** (%) | **Best Val. Error** (%) After fine-tuning for # epochs | | | |
|---|---|---|---|---|---|---|
| | | | 0 | 50 | 100 | 200 |
| − | Dense Baseline | 100.00 | 1.72 | | | |
| $\epsilon_g = 20\%$ | Constrained | 20.00 | 1.45 | | | |
| | Magnitude Pruning | − | 3.81 | 2.03 | 1.93 | 1.89 |
| $\epsilon_g = 10\%$ | Constrained | 10.02 | 1.51 | | | |
| | Magnitude Pruning | − | 9.31 | 2.63 | 2.57 | 2.48 |
| $\epsilon_g = 5\%$ | Constrained | 5.05 | 1.64 | | | |
| | Magnitude Pruning | − | 30.68 | 3.69 | 3.69 | 3.69 |
| $\epsilon_g = 1\%$ | Constrained | 2.62 | 1.92 | | | |
| | Magnitude Pruning | − | 90.45 | 60.82 | 60.69 | 55.45 |

Table 21: LeNet models trained with unstructured sparsity on MNIST. Unstructured magnitude pruning is applied independently at each layer to retain the desired target density. "Fine-tuning" for zero epochs means *no* fine-tuning. Metrics averaged across 3 runs.

| Target Density $g \in [1:4]$ | Method | $L_0$-**density** (%) | **Best Val. Error** (%) After fine-tuning for # epochs | | | |
|---|---|---|---|---|---|---|
| | | | 0 | 50 | 100 | 200 |
| − | Dense Baseline | 100.00 | 0.85 | | | |
| $\epsilon_g = 20\%$ | Constrained | 19.99 | 0.73 | | | |
| | Magnitude Pruning | − | 2.28 | 0.98 | 0.92 | 0.92 |
| $\epsilon_g = 10\%$ | Constrained | 10.00 | 0.78 | | | |
| | Magnitude Pruning | − | 5.23 | 1.38 | 1.38 | 1.32 |
| $\epsilon_g = 5\%$ | Constrained | 5.01 | 0.89 | | | |
| | Magnitude Pruning | − | 12.53 | 2.39 | 2.39 | 2.39 |
| $\epsilon_g = 1\%$ | Constrained | 1.55 | 1.26 | | | |
| | Magnitude Pruning | − | 88.76 | 88.76 | 88.76 | 88.76 |

**TinyImageNet.** Table 22 displays the result for TinyImageNet experiments at 1%, 5%, 10% and 20% unstructured sparsity. We observe similar patterns as in the MNIST experiments: the constrained approach reliably achieves the desired sparsity targets and preserves reasonable performance.

Table 22: ResNet18 models trained with unstructured sparsity on TinyImageNet. "Fine-tuning" for zero epochs means *no* fine-tuning.

| Target Density $g \in [1:16]$ | Method | $L_0$-**density** (%) | **Best Val. Error** (%) After fine-tuning for # epochs | | | |
|---|---|---|---|---|---|---|
| | | | 0 | 40 | 80 | 120 |
| − | Dense Baseline | 100.00 | 38.64 | | | |
| $\epsilon_g = 20\%$ | Constrained | 20.26 | 42.06 | | | |
| | Magnitude Pruning | − | 43.45 | 39.81 | 39.35 | 39.27 |
| $\epsilon_g = 10\%$ | Constrained | 10.57 | 42.54 | | | |
| | Magnitude Pruning | − | 54.06 | 42.19 | 41.45 | 41.25 |
| $\epsilon_g = 5\%$ | Constrained | 5.20 | 43.98 | | | |
| | Magnitude Pruning | − | 75.44 | 46.00 | 44.21 | 43.75 |
| $\epsilon_g = 1\%$ | Constrained | 1.87 | 47.24 | | | |
| | Magnitude Pruning | − | 99.21 | 81.67 | 72.95 | 69.00 |

In this task magnitude pruning outperforms the constrained approach, except for the case of 1% density. We hypothesize that the improvement of magnitude pruning at relatively larger density targets (10% and 20%) may be linked to the fact that the ResNet18 model is significantly larger than those models used for MNIST, and thus easier to fine-tune.

Finally, note that the performance of the constrained approach in the harsh 1% density setting is significantly better (albeit with a small violation of the constraint target) than that of magnitude pruning, even after fine-tuning the magnitude pruning model for 120 epochs.