# OpenReview forum: "Controlled Sparsity via Constrained Optimization or: How I Learned to Stop Tuning Penalties and Love Constraints"
_NeurIPS.cc/2022/Conference — NeurIPS 2022 Accept_

### Official Review · Reviewer_oV8j · 2022-07-08

**Rating:** 7
**Confidence:** 3
**Soundness:** 4 excellent
**Presentation:** 4 excellent
**Contribution:** 3 good

**Summary:**

The authors take the $L_0$ penalty method from Louizos et al. and present a constrained version of it, i.e., estimate the Lagrange multiplier to satisfy some desired level of parameter sparsity. They do this by specifying a desired parameter sparsity density, either for the entire model or per layer. The Lagrange factor appears to be estimated using (non-stochastic) gradient descent, with one additional trick: only apply the constraint when it isn’t satisfied. Experiments are performed using convolutional and fully-connected models on image datasets (CIFAR10, TinyImageNet, and ImageNet) in the supervised learning setting. They compare with the penalized version of the regularizer, as well as a pruning method based on weight magnitude.


**Questions:**

My main question is regarding the use of gradient descent for the Lagrange multiplier. Updating this factor once per epoch seems somewhat arbitrary, and I would guess that the issues with the slowly updating factor affecting training stem from this choice. Why not update the Lagrange multiplier online using stochastic gradients from each batch? Did you try this? The ‘dual restart heuristic’ may not be necessary in this case.

**Limitations:**

While the authors do not explicitly address the limitations of their work, they discuss the various tradeoffs of different sparsification algorithms to a reasonable extent.

**Strengths And Weaknesses:**

**Strengths**

*Originality*:
The originality of the paper is somewhat unclear to me, as I am not especially familiar with the latest techniques for neural network compression. To the best of my knowledge, I am not aware of previous works that allow setting a desired number of used parameters (rather than setting a Lagrange hyperparameter). In this regard, the paper presents a somewhat original method.

The authors also present a number of additional aspects, which includes a ‘dual restart heuristic’ and techniques for applying regularization to residual networks. These are mildly original, but I do not see them as a core contribution to the paper’s novelty.

*Quality*:
The paper is fairly high quality. The contributions of the paper are clearly defined. The diagrams clearly demonstrate the main benefit: being able to trace out the curve of parameters vs. performance in a more targeted manner. Experiments with multiple model architectures and datasets demonstrate the method, along with analysis plots in Figures 2 and 3.

*Clarity*:
The paper is very clear in its presentation. The writing is clear, containing emphases on important words and ideas. Mathematical concepts are generally presented and explained clearly. Experiments clearly demonstrate the settings described in the paper. The figures are all well-presented, with clear legends and labels.

*Significance*:
Assuming that the paper is at least moderately original, then I see the results as being at least somewhat significant.

In particular, in Table 2, the authors demonstrate that their method is able to train models that contain similar numbers of parameters as magnitude pruning while obtaining significantly improved validation accuracy (in some cases as 70% difference). In fairness, it would be reasonable to also compare with smaller models trained without weight regularization, as well as models that are optimized post-training. Thus, the significance may be somewhat diminished after these additional comparisons.

Again, assuming the technique has not been introduced previously, the notion of being able to pre-specify the size of the model could be useful, e.g., to ensure that the final model fits within memory on a smaller device.

The authors also claim that their method, along with several additional techniques discussed in Section 5.1, is able to train $L_0$-regularized residual networks, whereas, supposedly, standard $L_0$-regularization struggles in these settings. I am not familiar with this literature, so it’s somewhat unclear to me, but I see this as a further significant contribution. Many large-scale vision models contain residual components, and this set of techniques opens up the applicability of $L_0$-regularization to these networks.

**Weaknesses**

*Originality*:
The main contribution of the paper is to replace the $L_0$-regularized training objective with a constrained objective, adjusting the Lagrange factor to satisfy this constraint. While I’m generally in support of simple methods, this method is so simple as to make me question whether something similar hasn’t already been tried before. Replacing regularized objectives with their constrained counterparts is a common, classic technique within the machine learning literature. Some examples with modern deep networks can be found in variational autoencoders ([Rezende & Viola, 2018](https://arxiv.org/abs/1810.00597)) and policy regularization in reinforcement learning ([Haarnoja et al., 2018](https://arxiv.org/abs/1812.05905)). Given that this method is seemingly *so* simple, in my view, it should provide highly significant or surprising results to warrant publication.

*Quality*:
The quality of the paper is generally great, however, I have the following concerns:

It’s not clear whether post-training sparsification would obviate the issues with penalty-based regularization raised by the authors. That is, one could distill a larger, trained model into a smaller model of pre-specified size. In the introduction, the authors point to the additional computational overhead required by these methods, however, I do not find this to be the most compelling point. Additional discussion or empirical evaluation around this area would help to strengthen the authors’ claims.

Experiments are performed entirely within the setting of supervised learning for image classification. While it’s understandable to demonstrate a method with practical applications in domains that are commonly used in-practice, I worry that this limits the scope of the paper. A more complete paper would demonstrate this method in a range of settings (e.g., within various networks for generative modeling or reinforcement learning). Given the seeming simplicity of the method, it’s unclear why the authors did not demonstrate this technique more broadly.

*Clarity*:
I do not see any major weaknesses in the clarity of the paper. The main ideas are presented well.

*Significance*:
The significance of this paper may be somewhat limited.

As noted above, it may be the case that post-training sparsification methods would already provide a solution for obtaining a sparse network with a particular number of parameters. If this is the case, then the fact that this method is capable of doing so during training seems less consequential.

Also as mentioned above, the paper currently only contains empirical evaluations with supervised learning on image datasets. Expanding this to include other data modalities and tasks would expand the appeal of the paper.

The main selling-point of the paper is the ability to pre-specify the number of active parameters in the final, trained model. That is, one no longer has to manually adjust regularization penalty hyper-parameters during training. While I somewhat believe the appeal of this idea, in practice, I’m not sure whether this would actually save one from having to perform hyper-parameter tuning. The notion of having to pre-specify the density of active parameters (particularly per layer) is not so conceptually distinct from specifying the size (depth and width) of the network itself. Given that we generally sweep over these parameters, I do not see replacing a regularization penalty with a constraint to be a huge win here. Admittedly, this method allows one to set a global constraint, which is ideal compared with setting per-layer constraints / sizes.

The experiments in Table 2 (training ResNet50 on ImageNet) are fairly significant in terms of improvements in error. However, given the additional proposed techniques, I wonder whether a penalty-based regularization approach would perform similarly. If this is the case, then the only remaining benefit is the ease of tuning.

---

> ### Author Response · Authors · 2022-08-02
> **Response to Reviewer oV8j**
>
> Thank you for your review. We would like to bring some clarifications related to the raised questions and concerns.
>
> ### In- vs post-training sparsity.
>
> The reviewer challenges the motivation of "in-training" sparsification, as opposed to "post-training" pruning of a densely-trained model. The debate on whether to perform "in-" vs "post-training" sparsification is a very interesting question, and central to much of the research taking place in the current sparsity community. While we believe further exploration on the trade-offs (regarding performance, computational cost, data availability) between in- and post-training quantization is needed, answering this questions is beyond the scope of our work.
>
>
> ### Additional applications
>
> The reviewer suggests to demonstrate our technique in other domains, such as generative models and reinforcement learning. Undoubtedly, these experiments would lead to a more complete publication. However, the standard practice in the current sparsity literature is to concentrate on supervised learning tasks to validate new methods and progress.    For this reason, we decided on a similar evaluation setup. This being said, we see our positive results as an invitation to the community interested in problems _beyond supervised learning_ to apply our proposed method in their respective areas.
>
> ### In-training sparsity vs architecture search
>
> In the context of architecture search, we agree with the reviewer that there is a conceptual similarity between determining the size of a layer/model and determining the sparsity level. However, there is a notable practical difference: performing in-training sparsity allows for greater flexiblity during training since it allows for **exploring multiple "architecture configurations" during optimization**. In contrast, using *fixed, manually-selected* architectures, would require a large number of experiments to explore the "combinatorial" space of model size configurations.
>
> ### Constrained vs penalized
> >  "I wonder whether a penalty-based regularization approach would perform similarly. If this is the case, then the only remaining benefit is the ease of tuning."
>
> We would like to remind the reviewer that the advantages of our method go beyond "ease of tuning". In practice, adjusting the penalty hyperparameter might be prohibitively expensive for large-scale tasks. We demonstrate that this challenge is present even for simple MNIST tasks with MLP models on Appendix E.
>
> Moreover, Boyd and Vandenberghe (Sec. 4.7, 2004) discuss how the constrained approach can be strictly more powerful than the penalized method: when using the penalized formulation, **there may exist values of the target density that cannot be achieved for any value of the penalty parameter!** This means that tuning the hyperparameter to achieve said density level can be altogether pointless, regardless of the amount of computational resources available.
>
> ### Significance
>
> Despite the simplicity of our proposal, our work successfully addresses a practical challenge in the sparsity literature for training residual models with L0-regularization. Moreover, our submission presents a solution to this issue **and** our proposal grants improved tunability and hyperparameter-interpretability without compromising on performance. As pointed out by Reviewer TJdj:
> > "I think the results of this work are impactful. L0-regularization was a very promising technique but prior work (Gale et al.) was unable to scale it to large models. The authors addressed its limitations both in the difficulty of tuning the loss coefficient to achieve a desired level of sparsity and the previously reported issues with training stability."
>
> ---
>
> ### Q1 - Timing on updates to Lagrange multipliers
>
> The Lagrange multipliers are *not* updated once every epoch. **We update the multipliers at every step (i.e. at every batch).** Note that the updates to the multipliers are *not* stochastic since the expected L0-norm depends only on the model parameters, and *not* on the sampled minibatch.
>
> ---
>
> ### References:
> - S. Boyd and L. Vandenberghe. Convex Optimization. Cambridge University Press, 2004.

---

> > ### Comment · Reviewer_oV8j · 2022-08-09
> > **Response to Authors**
> >
> > Thank you for your in-depth replies to me and the other reviewers. After reading the reviews and responses, I am convinced that the paper presents a compelling idea along with supporting experiments. Accordingly, I have decided to increase my score to 7 / Accept.

---

### Official Review · Reviewer_ep34 · 2022-07-08

**Rating:** 8
**Confidence:** 4
**Soundness:** 3 good
**Presentation:** 3 good
**Contribution:** 3 good

**Summary:**

This paper considers sparse neural network training with cardinality constraints instead of regularization. Based on the stochastic gates, they adopted a constrained formulation, which allows for simultaneously training and controlling exact sparsity in an end-to-end fashion. In terms of optimization algorithm, they considered the Lagrangian min-max problem and proposed gradient-descent-ascent algorithm with dual restarts. They conducted extensive and comprehensive numerical experiments on various network models, and demonstrated the effectiveness of their proposal.

**Questions:**

See "Strengths And Weaknesses"

**Ethics Review Area:**

["I don’t know"]

**Limitations:**

See "Strengths And Weaknesses"

**Strengths And Weaknesses:**

The paper proposes an end-to-end training framework for neural network training with exact sparsity constraint, which avoids tuning the sparsity-inducing regularization coefficients to satisfy the desired sparsity. They extended the previous work on stochastic gates (Louizos et al., 2018) to the constrained version, and solve it by gradient descent ascent on its Lagrangian. The paper is well written and comprehensive. Their extensive numerical experiments are sound and demonstrate their method consistently help achieve the sparsity targets for various models. In general, the paper is solid and good.
Here are some comments/questions:

1. Line 91-94: the introduction of $\phi$ is a bit abrupt and may not be friendly to those who have not read the previous work Louizos et al. 2018. It is probably better to describe the idea of Louizos’ paper, and provide explicit expression of $\lambda_{\text{pen}}\mathbb{E}_{\bf{z}|\phi}[\|z\|_0]$ wrt $\phi$.
2. The $L_0$-density throughout the paper (in experiment section, figures and tables): is it the “smoothed” version $g_{\text{const}}(\phi_g)$ (the LHS of the constraint) or the density based on test-time model? It is sort of confusing. At first, I thought it was the actual parameter density of the final purged model. So, it is probably better to clarify/define the term somewhere.

3. Question related to test-time and purging: each $\mathbb{E}_{\bf{z}|\phi}[1\{z_i\neq 0\}]$ is a number between $[0,1]$. This means even $g (\phi_g)\leq \epsilon_g$ holds, there are still chances that the $z$ in the test-time model has a larger density compared to $\epsilon_g$. But why the Params(%) is in general smaller than $L_0$-density in Table 2?
4. Question related to $L_0$-density: in the left panel of Figure 3, the curves of $L_0$-density do not meet the target sparsity level?  Any intuition why that happens?
5. As a following work of Louizos et al. 2018, Yutaro et al. 2020 [1] mentioned the logistic-based hard-concrete distribution (also used in this paper) “yields high-variance gradient estimates”, and they proposed to use Gaussian distribution to replace the HC distribution for $z$. It would be interesting to check if Gaussian distribution works better in your case.

[1] Yamada, Yutaro, Ofir Lindenbaum, Sahand Negahban, and Yuval Kluger. "Feature selection using stochastic gates." In International Conference on Machine Learning, pp. 10648-10659. PMLR, 2020.

---

> ### Author Response · Authors · 2022-08-02
> **Response to Reviewer ep34**
>
> We appreciate the reviewer's careful reading of our submission.
>
> ### Q1 - Introduction of $\phi$
> We agree with the reviewer's recommendation. In fact, this brief presentation of Louizos' previous work is already contained in Appendix A of our submission. Due to space constraints, we were unable to include this in the main body of the paper.
>
> Unfortunately, we had not included a pointer to this appendix in the submission. We have fixed this in the uploaded revision (lines 92-94), and hope that the reviewer will find this strategy a reasonable compromise.
>
> ---
>
> ### Q2 - Clarifying the meaning of $L_0$-density
> Whenever we mention the $L_0$-density, we indeed mean the "smoothed"  version (i.e. the expected, normalized $L_0$-norm) $g\_{\text{const}}(\phi_g) = \frac{\mathbb{E}\_{z_g | \phi_g} \\left[ ||z_g||_0 \\right]}{\\#(\tilde{\theta}_g)}$. When we report the parameters and MAC counts, these correspond to the final purged models.
>
> We have added clarifications to resolve this potential confusion in the revised version:
> * Caption of Fig 1
> * Overbrace annotation on Eq 3
> * Adjusted lines 249-250
> * Made all figure labels consistent when referring to $L_0$-density
>
> ---
>
> ### Q3 - Test-time and purging
>
> We agree with all your mathematical statements in the preamble to your question.
>
> When training models with adjacent sparsifiable modules/layers, the sparsification effect "compounds". This is the reason behind the lower number of test-time active parameters compared to the $L_0$-density of the model. We refer to this behavior as "double-sparsification".
>
> **Please note that our submission contains a discussion of this subtle phenomenon in Appendix D.**
>
>
> The following link points to an (anonymized) annotated version of Figure 11 from our appendix. Note that the discussion below can be understood without viewing the annotated figure. https://imgur.com/a/371mjJu
>
>
> Suppose both layer k and k+1 drop 50% of their output neurons. Then layer k+1 can also drop 50% of its **inputs** (those units dropped by layer k!). Therefore, although each layer might have a 50% target density, the resulting sparsity of the purged model can be lower. This "double sparsification" phenomenon is more pronounced when training with layer-wise, rather than model-wise, constraints.
>
> Here, we chose to illustrate this behavior with MLP models because the trend is easier to visualize. However, this phenomenon is present for any model with back-to-back sparsifiable layers. In particular, this includes our trained ResNet models.
>
> Note that this behavior is less likely to occur in the context of unstructured sparsity since it would require *all the individual connections* arriving or departing from a neuron to be turned off. If at least one such connection is on, the neuron would not be discarded.
>
>
> **We highlight that this phenomenon stems from the use of structured sparsity, and is not due to the choice of a L0-reparametrization or to the use of our constrained formulation**. Addressing this issue (and its potential benefits or limitations) is not the focus of our work.
>
> ---
>
> ### Q4 - Density target not met in Fig3
>
> Thank you for spotting this! The TinyImagenet + ResNet18 + model-wise target density of 0.2 experiment was improperly tuned. We have re-ran this experiment with an appropriate configuration and updated Figure 3.
>
> Note that (as described in Appendix J.6) in our TinyImageNet experiments we use **the same hyperparameters** across all the different target densities. A priori, one could adjust the hyperparameters for each individual target density. We decided in favor of using a unique hyperparameter choice as a way to demonstrate the robustness of our method.
>
> This meant that the chosen hyperparameters worked fine for all other examined target densities except 0.2. Our updated experiments resolve this issue.
>
> ---
>
> ### Q5 - Work of Yamada et al. (2020)
>
> Thank you for bringing this interesting work to our attention.
>
> Our choice for the L0-reparametrization of Louizos et al. (2018) was based on three factors: 1) its wide adoption in the ML sparsity literature; 2) its simplicity for adapting into the constrained framework and 3) the practical challenges documented by Gale et al. (2019).
>
> In principle, changing the distribution of the stochastic gates should integrate seamlessly with our proposed constrained formulation. According to Appendix 3 of Yamada et al. (2020), this new distribution seems to satisfy the conditions on efficient computation and differentiability for the expected L0-norm.
>
> Studying other reparametrizations with better optimization properties (such as the lower gradient variance of Yamada et al. (2020)) is an intriguing direction for future work.

---

### Official Review · Reviewer_syaD · 2022-07-11

**Rating:** 5
**Confidence:** 4
**Soundness:** 3 good
**Presentation:** 3 good
**Contribution:** 2 fair

**Summary:**

This paper mainly focuses on a previous work l0 regularization and demonstrate its feasibility  when solving it with constraints. The work considers the problem of L0 regularization thoroughly by 1) learning models with controllable levels of sparsity 2) dual restart heuristic to avoid the excessive regularization 3) fixing the performance dropping problem of L0 regularization.

**Questions:**

1. Can you present more comparisons with more sota methods and sparser region [2] ?

2. Can this method be applied to sparse training and improve performance?

[2] Winning the Lottery with Continuous Sparsification

**Ethics Review Area:**

["I don’t know"]

**Limitations:**

I don't see any negative societal impact of this work.

**Strengths And Weaknesses:**

Strengths:
1. This work presents valid improvements to previous L0 regularization, since I know L0 regularization cannot achieve stable results on ImageNet.

2. This work gives a nice modification to the original test-time model selection criterion for its mediean property.

Weaknesses:
1. There is already work on dealing with the penality and unstable problem of L0 regularzation [1](Effective Sparsification of Neural Networks with Global Sparsity Constraint) and the final accuracy result is lower than [1] on ImageNet (73.44\% with 12% params vs. 74.68% with 10% params). Please discuss the difference and connections with [1] and better make more empirical comparisons.

2. The good property of test-time model selection should be further justified. Can you please give a table of statistics to demonstrate the changing dynamics of the test-time model selected during the training process?

---

> ### Author Response · Authors · 2022-08-02
> **Response to Reviewer syaD**
>
> ### W1 - Previous work of Zhou et al. (2021)
>
> Thank you for bringing this highly relevant work to our attention. We have complemented our related work section (lines 212-214) to reference this work explicitly. Below, we discuss in more detail the similarities and differences between the two approaches.
>
> **Summary of Zhou et al. (2021)**
>
> Zhou et al. (2021) consider the constrained optimization problem (adjusted to our notation):
> $\min\_{\tilde{\theta}, z}  \mathcal{L}_{\mathcal{D}}(\theta = \tilde{\theta} \odot z) \text{ s.t. } ||z||_1 \le K \text{ and } z \in \\{0, 1\\}^{|\theta|}$
>
> This problem is then relaxed by means of a stochastic reparametrization into:
> $\min_{\tilde{\theta}, z} \mathbb{E}\_{z|s} \mathcal{L}\_{\mathcal{D}}(\tilde{\theta} \odot z) \text{ s.t. } ||s||_1 \le K \text{ and } s \in [0, 1]^{|\theta|}$
>
> The authors employ the Gumbel trick to estimate the gradients of their objective function with respect to the mask learnable parameters $s$. Moreover, they cleverly exploit the existence of an efficient projection step to their feasible set.
>
> **- Discussion -**
>
> Although both works consider a constrained optimization perspective on sparsity, there are crucial differences regarding the focus of the papers, the adopted solution strategies, and additional requirements/assumptions.
>
> **Performance comparison:** The reviewer suggests a performance comparison based on the results reported by Zhou et al. (2021). Unfortunately, their experiments were carried out in the unstructured sparsity setting, while our experiments focused on structured sparsity.
>
> Considering the large differences in acceleration (computation savings) and the compression rates (storage savings) between structured and unstructured sparsity methods, comparing to their reported results would be inadequate.
>
> **Focus:** We make a conscious effort to illustrate the shortcomings of the penalty-based approach, and how the constraint formulation alleviates/solves some of these issues. In particular, our work concentrates on the aspects of tunability and hyperparameter interpretability.
>
>
> **Solving the constrained problem:** The use of projected gradient descent in Zhou et al. (2021) obviates the need for explicitly optimizing the Lagrange multipliers. In contrast, we tackle the associated min-max Lagrangian problem.
>
> Performing the projection step exactly can be appealing since it can guarantee a desired level of sparsity is met throughout training. However, the authors apply projected gradient descent on a relaxation of the original problem. In other words, they *exactly* solve an *approximate* problem.
>
> Despite the risk of unstable/oscillatory behavior associated with the Lagrangian problem, our experiments using simple gradient descent-ascent (with our proposed dual restarts scheme) exhibited stable and successful training across many datasets and architectures.
>
>
> **Extensibility:** A key advantage of our constrained proposal is its modularity/extensibility: new requirements can be directly incorporated as additional constraints. Formulating these constraints is certainly possible in the framework of Zhou et al. (2021). However, their use of projected gradient descent limits their applicability to constraints with an efficiently-computable projection operator. In contrast, our Lagrangian formulation can operate on any differentiable constraints (note that we do not aim to provide convergence guarantees!).
>
> **Additional hyperparameters:** The algorithm presented by Zhou et al. (2021) requires the use of a "precise[ly] chosen temperature annealing scheme". The authors also employ a gradually increasing pruning rate "to make a smooth transition from dense to sparse status". These two additional schedule hyperparameters are not present in our work.
>
> ---
>
> ### W2 - Dynamics of test-time model
>
> We provide an extensive discussion of the dynamics of the gate medians (i.e. those used for the test-time models) in Appendices H and I. In particular, we study how the gate dynamics are affected by the choice of learning rate for the gates and the use of weight decay.
>
> Please let us know if you have any specific questions that are not addressed in Appendices H and I.
>
> ---
>
> ### Q1 - Comparison to SOTA methods and sparser region
>
> Please see our responses to "Q2 - Higher sparsity levels" and "Q3 - Q3 - Comparison to other SOTA sparsity methods" for Reviewer TJdj.
>
> ---
>
> ### Q2 - Better performance when applied for sparse training?
>
> Applying our method to ResNet models resulted in significant performance gains compared to the challenges for training residual models with the L0 reparametrization reported by Gale et al. (2019).  We hope our results, along with the simplicity of the proposed approach, will motivate future works to investigate the potential of this method to improve performance across a wide range of applications.

---

### Official Review · Reviewer_TJdj · 2022-07-11

**Rating:** 7
**Confidence:** 4
**Soundness:** 4 excellent
**Presentation:** 4 excellent
**Contribution:** 3 good

**Summary:**

The authors propose a new approach for enforcing sparsity with the L0-regularization technique proposed by Louizos et al. that allows for explicit specification of the target sparsity at the end of the optimization process. They demonstrate that their approach produces high quality sparse models on a range of applications and datasets.


**Questions:**

1. I wonder whether your fixes for the training instability reported by Gale et al. would generalize to unstructured sparsity as they explored. If you could comment on this or include results in the paper I think it would be useful to readers.
2. In Table 2 it would be nice to include some higher sparsity levels. I recognize that you’re inducing structured sparsity and that expected compression levels will be lower than with unstructured but I think it would be useful to readers to see up to 50% compression.
3. If you wanted to demonstrate another level of impact you could compare your improved L0-regularization technique to other state-of-the-art methods for neuron/channel pruning. I do not think this is necessary for this work to be published at NeurIPS, but it would be a way to increase the impact of the contributions further.


**Limitations:**

I am not aware of any limitations of the proposed approach that were not mentioned in the work. The authors did explicitly analyze potential issues - for example, the dynamics of their constraint during training.

**Strengths And Weaknesses:**

Strengths
1. The paper is very well written and easy to follow. The authors do a good job of highlighting the benefits of their technique, including the ability to target specific sparsity levels and the modularity and extensibility provided by this capability.
2. I think the results of this work are impactful. L0-regularization was a very promising technique but prior work (Gale et al.) was unable to scale it to large models. The authors addressed its limitations both in the difficulty of tuning the loss coefficient to achieve a desired level of sparsity and the previously reported issues with training stability

I did not identify any major weaknesses. In the next section I highlight a few things that could help to bring my score up further.

---

> ### Author Response · Authors · 2022-08-02
> **Response to Reviewer TJdj**
>
> Thank you for your encouraging comments on our work. Please see individual responses to your questions below:
>
> ### Q1 - Unstructured sparsity
> We share the reviewer's curiosity to evaluate the performance of our approach in the unstructured sparsity setting. During the response period we have modified our code to support unstructured sparsity tasks (i.e. one gate for each individual weight and bias).
>
> Due to time constraints we do not yet have ResNet50-ImageNet that would allow us to directly compare with the work of Gale et al. (2019). However, we have preliminary results on applying our method to a ResNet18 network on TinyImageNet. In this experiment **we were able to obtain a model with 80% unstructured sparsity (20% density), while retaining a validation error of ~43%.** This performance is in a similar range to that of Table 16 in Appendix K4 for the structured case.
>
> Note that Gale et al. (2019) report that in their experiments whenever the residual model became sparse, the validation accuracy behaved like random guessing. In that respect, our preliminary results seem very promising.
>
> We intend to carry out more comprehensive experiments in the unstructured regime for the camera-ready version of the paper.
>
> ---
>
> ### Q2 - Higher sparsity levels
> As requested, we have carried out experiments on ImageNet with lower target densities (30% and 50%). We have updated our manuscript to include results with 30%, 50%, 70% and 90% (structured) density targets. For convenience we present a summarized version of the table below.
>
> We reliably achieve the sparsity target in all cases, including the new harsher sparsity regimes. At 70% and 90% density we are competitive with magnitude pruning (after fine-tuning the magnitude pruning network for 20 epochs). Moreover, **at lower densities (30% and 50%) our approach delivers significant performance improvements compared to magnitude pruning with fine-tuning.**
>
> The results below confirm that model-wise constraints result in better performance than layer-wise constraints. This is due to the fact that the layer-wise constraints demand a uniform sparsity across the whole model, while the model-wise constraint enables the model to allocate sparsity depending on the "relevance" of the layer. Note that the model-wise setting also produces models higher MAC counts.
>
> | Target Density (%) | Method |  L0 -density (%) | Params (%) | MACs (%) | Val. Error (%) | Val. Error w/ Fine-Tuning (%) |
> | -------- | -------- | -------- | -------- | -------- | -------- | -------- |
> | N/A | Pre-trained Baseline |  100  | [25.5M] | [4.12 x 10^9] |  23.90  | - |
> | 90     | Model-wise     |  90.36      |   88.06      |  91.62      |  24.68      | -     |
> | 90     | Layer-wise     |  90.58      |   87.07      |  85.97      |  24.97      | -     |
> | 90     | Layer-L1 Mag. Prune     | -     |   85.94      |  84.99      |  38.74      |  24.68      |
> | 70     | Model-wise     |  70.78      |   64.41      |  76.50      |  25.53      | -     |
> | 70     | Layer-wise     |  70.36      |   61.91      |  58.59      |  26.98      | -     |
> | 70     | Layer-L1 Mag. Prune     | -     |   62.15      |  59.85      |   97.78     |    26.14   |
> | 50     | Model-wise     |  50.18      |   42.47      |  58.00      |  27.51      | -     |
> | 50     | Layer-wise     |  50.70      |   43.15      |  38.25      |  27.89      | -     |
> |  50      | Layer-L1 Mag. Prune     | -     |   43.47      |  39.76      |  99.75      |  29.16      |
> |  30      | Model-wise     |  30.31      |   31.81      |  42.05      |  29.65      | -     |
> |  30      | Layer-wise |  31.44      |   30.16      |  23.74      |  31.71      | -     |
> |  30      | Layer-L1 Mag. Prune    | -     |   29.86      |  24.80      |  99.89      |  34.74      |
>
> ---
>
> ### Q3 - Comparison to other SOTA sparsity methods
> We appreciate the reviewer's assessment of sufficiency of our current contribution.
>
> A core objective of our work was to demonstrate that general-purpose constrained optimization methods can be successful at handling sparsity tasks with modern architectures and datasets, **and** simultaneously provide improved interpretability and modularity/extensibility compared to the penalized approach.
>
> Our results demonstrate that the adoption of constrained formulations is a competitive strategy for sparse training. We hope that our work will motivate future efforts towards improving performance using constrained methods.

---

> > ### Comment · Reviewer_TJdj · 2022-08-05
> > **Reviewer response**
> >
> > Thank you to the authors for the responses. I agree that the preliminary results on unstructured sparsity look promising compared to the results reported by Gale et al. The updates results at higher sparsity levels are also compelling and I think they make the paper more comprehensive. As I said in my initial review, I think this is a good paper that should be accepted.

---

### Author Response · Authors · 2022-08-08
**Clarification on updated ImageNet results**

During the rebuttal period, reviewer TJdj suggested carrying out ImageNet experiments on a sparser regime. We took advantage of this additional batch of experiments to tune the initialization scheme for the learnable parameters of the stochastic gates.

Initially, we performed our ImageNet experiments using the value of the initialization hyperparameter suggested by Louizos et al. (2018) in their WideResNet experiments ($\rho_{\text{init}}=0.3$). This hyperparameter has an inverse relation with the active probability of the gates at initialization. We identified that using $\rho_{\text{init}}=0.3$ leads to an initial sparsity configuration that is quite restrictive compared to the fully dense version of the model. During our tuning, we found that lowering the value of $\rho_{\text{init}}$ to $0.05$ consistently improved validation performance due to the recovered expressive capacity of the model at initialization.

The ImageNet results reported in the revised version of the paper employ $\rho_{\text{init}}=0.05$. Please see Appendices A.2 *"Initialization of the gates"* and J.7 *"Experimental Details > ImageNet"* for a more detailed discussion.

---

### Meta-Review · Area_Chair_7wYQ · 2022-08-24

**Recommendation:** Accept
**Confidence:** Certain

**Metareview:**

Ratings: 7/8/5/7.
Confidence: 3/4/4/4.
Discussion among reviewers: No.

Summary: this paper introduces a method for learning neural networks with an exact sparsity % target instead of regularization constraints. The method builds on the smoothed L0 regularization objective from Louizos et al, which was shown to be difficult to scale.

The reviewers generally agree that the paper is easy to follow, introduces ideas that are interesting to the NeurIPS community, and that the results look promising.

The authors wrote detailed responses to the reviewers' concerns. During the rebuttal period, the authors performed additional experiments on ImageNet as suggested by reviewer TJdj, and updated their paper. The reviewer(s) did not respond to this update, but their reviews were already positive. My recommendation is to accept.

**Award:**

No

---

### Decision · Program_Chairs · 2022-09-14

Accept